# A Neuroscience-inspired Framework for Tri-modality Alignment of Brain Signals, Vision, and Language

## Abstract

Visual retrieval from brain signals is a key challenge in Brain-Computer Interfaces (BCIs). Existing methods mainly rely on direct cross-modality mapping, yet they often overlook the neural mechanisms of visual processing, which leads to three major limitations. First, a feature physiology mismatch arises because high-level semantic features extracted by image encoders do not align with the low-level neural responses evoked by rapid visual stimulation. Second, most approaches emphasize cross-modality alignment while neglecting the similarity of neural representations within the same category, which results in poor intra-modality semantic consistency. Third, brain image alignment typically depends on static image text semantic spaces and therefore lacks dynamic semantic priors that interact with brain activity. We introduce NeuroAlign, the neuroscience-inspired framework for brain visual alignment. NeuroAlign mitigates the feature physiology mismatch by integrating bottom-up structural perception with top-down semantic modulation, enhances semantic consistency through intra-modality self-supervision and cross-modality intra-class constraints, and leverages large language models (LLMs) to provide dynamic semantic signals that interact bidirectionally with brain responses. Extensive experiments demonstrate that NeuroAlign achieves state-of-the-art performance on both intra-subject and inter-subject retrieval tasks, which validates the effectiveness of this neuroscience-inspired alignment strategy.

## 1 Introduction

When humans observe an image, the brain can recognize complex visual scenes within approximately 300 ms (Thorpe et al., 1996; DiCarlo & Cox, 2007; Cichy et al., 2014). In contrast, decoding visual stimuli from electroencephalography (EEG) signals, whether for image retrieval (Du et al., 2023; Song et al., 2024; Li et al., 2024a; Song et al., 2025; Wu et al., 2025), image reconstruction (Takagi & Nishimoto, 2023; Scotti et al., 2023; 2024; Ma et al., 2025), or imagination reconstruction (Shimizu & Srinivasan, 2022; Koide-Majima et al., 2024), still falls far behind human-level perception. This gap raises a central question: **Do existing methods overlook critical mechanisms of visual processing in our brain?** Despite the rapid progress enabled by deep learning, the representations captured from neural signals remain fundamentally misaligned with the brains actual processing dynamics.

Neuroscience has revealed that the visual system is not a passive feedforward processor. Instead, perception emerges from the *dynamic interplay* between bottom-up sensory input and top-down prior expectations (Desimone et al., 1995; Corbetta & Shulman, 2002; Buschman & Miller, 2007). As shown in Figure 1, the human visual system exhibits a dual-stream architecture (Li et al., 2025a): Signals from the retina are first processed in V1 cortex for initial processing and then passed to higher-order cortices for semantic abstraction, where bottom-up stimulus-driven attention and top-down task-driven attention interact dynamically. However, most current brain-to-image decoding methods adopt a direct cross-modality mapping strategy by aligning neural features with representations extracted by pretrained vision encoders such as Deep Residual Networks (ResNet) (He et al., 2016) and Contrastive Language-Image Pre-training (CLIP) (Radford et al., 2021) (Palazzo et al., 2020; Ye et al., 2024). Some works like (Choi et al., 2023) are also inspired by the dual-stream model, but they focus on encoding fMRI signals to model visual behavior, while ours is on decoding

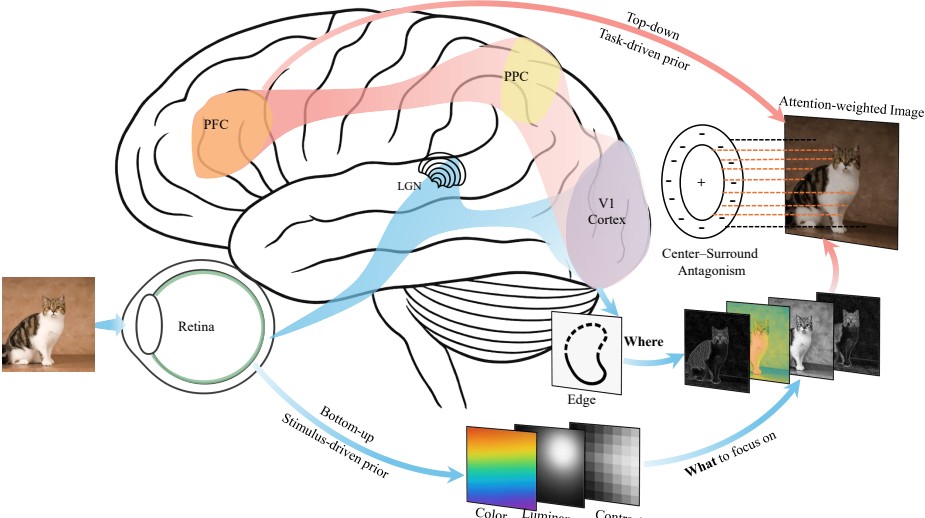

Figure 1: Neuroscience mechanism of brain visual and language processing. The blue pathway indicates bottom-up stimulus-driven attention: signals originate from the retina, pass through the lateral geniculate nucleus (LGN), and transmits visual information to the primary visual cortex (V1). The red pathway indicates top-down task-driven attention: the posterior parietal cortex (PPC) is responsible for integrating information from V1, while the prefrontal cortex (PFC) supports top-down attentional control by providing semantic guidance. The NeuroAlign framework simulates this dual-stream parallel processing mechanism.

EEG/MEG signals for image retrieval, which presents unique challenges due to the lower signal-to-noise (SNR) ratio and high temporal dynamics of EEG signals. This paradigm fundamentally diverges from the brains true processing mechanism and leads to three major bottlenecks:

**Bottleneck 1: Feature-physiology misalignment.** Conventional vision encoders are optimized for high-level semantic abstraction, with deep features corresponding to late-stage cortical processing (Cichy et al., 2016). Yet, under rapid serial visual presentation (RSVP) paradigms, neural responses are more sensitive to low-level cues such as edges, luminance, and color (Hubel & Wiesel, 1968; Itti & Koch, 2000). This results in a significant mismatch between the image features and the actual brain signals.

**Bottleneck 2: Lack of intra-modality semantic consistency.** Neuroscience studies suggest that stimuli from the same semantic category should evoke similar neural representations (Kriegeskorte et al., 2008; Cichy et al., 2014), providing the basis for robust recognition. Yet, due to the inherent low SNR of EEG, existing contrastive approaches (e.g., CLIP) that mainly enforce instance-level alignment are insufficient. Without explicit guidance from a well-structured semantic distribution, they fail to guide the same category EEG samples to aggregate effectively. Consequently, the learned representations remain distinctively weak, leading to a lack of intra-modality consistency (Liang et al., 2022; Mistretta et al., 2025; Tao et al., 2025) and reducing decoding generalization.

**Bottleneck 3: Static and unidirectional semantic priors.** Human perception relies on the bidirectional interaction of bottom-up sensory information and top-down semantic priors (Corbetta & Shulman, 2002; Bar et al., 2006; DiCarlo & Cox, 2007; Chiou & Ralph, 2016; Kar et al., 2019). However, existing brain-vision alignment approaches rarely leverage textual priors effectively (Song et al., 2024; Li et al., 2024a; Wu et al., 2025). Some methods incorporate text in a two-stage fashion (first aligning image and text, and then aligning image and brain), which forces brain signals to fit into a vision-centric space without exploiting the dynamic guiding role of semantic priors (Song et al., 2025). This unidirectional approach lacks sufficient top-down modulation, a key feature that aligns visual processing with semantic priors in the human brain.

To address these limitations, we propose **NeuroAlign**, a neuroscience-inspired framework for brain-vision alignment that integrates the language modality as a dynamic guiding signal. Our contributions are as follows:

- We design a visual saliency extraction module that captures RSVP-relevant cues (edges, luminance, contrast, color), enabling more biologically grounded alignment between visual and neural features.

- We employ large language models (LLMs) to generate image descriptions and leverage KL divergence for constructing cross-modality semantic guidance, optimizing the semantic consistency in the joint brain-vision-language feature space.

- We simulate the brains bidirectional interaction between bottom-up stimuli and top-down priors, where semantic guidance actively aligns neural signals with salient visual features instead of passively fitting them to image encoder features.

- Through large-scale evaluation on the THINGS-EEG2 dataset, NeuroAlign achieves state-of-the-art performance on 200-class zero-shot retrieval: **48.1% Top-1** and **78.1% Top-5** in within-subject evaluation, and **14.5% Top-1** and **36.4% Top-5** in cross-subject evaluation, substantially outperforming prior methods and demonstrating superior generalization.

## 2 RELATED WORKS

### 2.1 VISUAL BRAIN DECODING

Visual brain decoding aims to recover or retrieve visual-related appearance information (such as shape, color, texture) and semantic information (such as object categories, scene meanings) from human neural activity (Lin et al., 2022; Takagi & Nishimoto, 2023; Spampinato et al., 2017; Gaziv et al., 2022). In recent years, electroencephalography (EEG) has become a mainstream carrier for capturing neural activity due to its millisecond-level temporal resolution and portability (Liu et al., 2025). However, EEG suffers from problems of low signal-to-noise ratio and spatial resolution (Li et al., 2025b), making the decoding of visual semantic information from EEG signals heavily dependent on integrating multimodal data (Li et al., 2024b). To better understand the visual processing mechanisms of the human brain under natural conditions, researchers have adopted the Rapid Serial Visual Presentation (RSVP) paradigm (Intraub, 1981; Keysers et al., 2001; Gifford et al., 2022), which can simulate continuous, rapidly changing visual inputs in real environments, thereby revealing the temporal dynamic characteristics of brain visual information processing. Based on the RSVP paradigm, researchers have constructed large-scale neural datasets (Grootswagers et al., 2019; Gifford et al., 2022; Hebart et al., 2023), and conducted a series of visual decoding studies on this foundation (Benchetrit et al., 2024; Liu et al., 2024a). Although these works have achieved significant progress, current methods still largely treat the brain as a "black box" to a great extent, ignoring the intrinsic neural mechanisms of the brain during rapid visual processing. Therefore, combining neuroscience theory with deep learning decoding models is a key direction for improving decoding performance under the RSVP paradigm.

### 2.2 CROSS-MODALITY CONTRASTIVE LEARNING

In the field of cross-modality learning, CLIP effectively optimizes multimodal joint representations and enables zero-shot knowledge transfer by aligning the feature distributions of language and vision modalities (Li et al., 2024b). This approach has been introduced into neural decoding, where neural signals are aligned with external visual stimuli within a shared multimodal embedding space to perform downstream tasks such as classification, retrieval, or reconstruction (Bai et al., 2024; Choi & Ishikawa, 2024; Chen et al., 2024; Yang & Liu, 2024). However, the core contrastive loss in CLIP focuses on the similarity between cross-modality paired samples, failing to impose explicit constraints on intra-modality semantics (Liang et al., 2022; Wang et al., 2025). This leads to a lack of semantic consistency in the embedding space; for instance, semantically similar samples of the same visual category might be mapped to distant locations (Mistretta et al., 2025), which contradicts the human approach to classification and semantic understanding (Huth et al., 2016). To mitigate this issue, recent studies have attempted to introduce external priors to enhance semantic consistency, such as using Large Language Models to generate fine-grained image descriptions (Song et al.,

2025), or employing clustering and prototype learning to aggregate text features (Li et al., 2024b). Similar applications also exist for high-spatial-resolution fMRI, which relies on Representational Similarity Matrices (RSM) to align its global topology (Zhou et al., 2024). Nevertheless, these improvements remain dominated by text semantics, statically aligning brain signals to the image-text space. They have failed to effectively mine the inherent semantic information within EEG signals and have overlooked the representational differences between modalities due to direct alignment, leaving significant room for improvement in decoding accuracy and cross-subject generalization capability.

## 3 METHODS

To address the three bottlenecks identified, feature-physiology mismatch, lack of intra-modality semantic consistency, and static/unidirectional semantic priors, we propose NeuroAlign, a neuroscience-inspired dynamic semantic alignment framework. The overall architecture is illustrated in Figure 2.

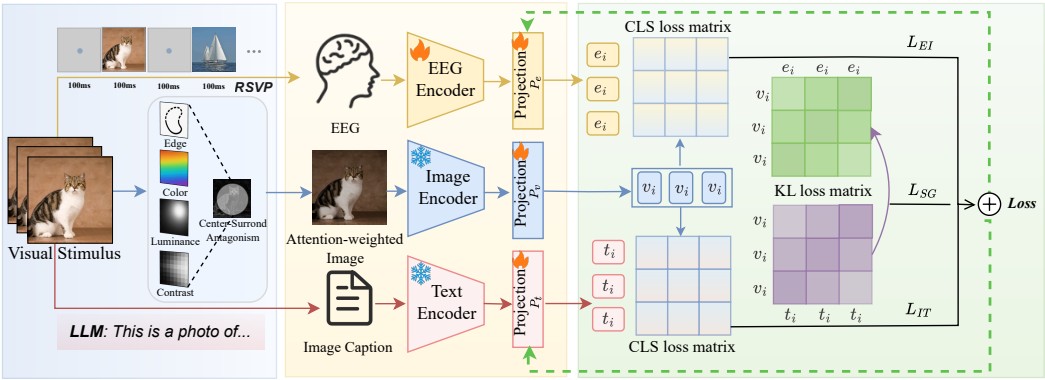

Figure 2: Overview of the NeuroAlign Framework. 1) Simulates the early visual processing of the human brain to extract salient features; 2) Establishes image-text supervision through contrastive learning with intra-modality loss, guiding EEG-image alignment based on KL divergence; 3) Mimics attention allocation in the brain's cognitive process to optimize semantic-guided EEG-image alignment.

### 3.1 OVERVIEW

NeuroAlign combines bottom-up EEG feature encoding with top-down semantic guidance to achieve robust cross-modality alignment. The framework consists of three core components: 1) A Visual Saliency Extraction (VSE) module that mimics early visual processing to extract physiologically relevant features from images; 2) A Semantic Guidance Alignment (SGA) module that uses image-text semantic priors to guide EEG-image representation learning; 3) A Dynamic Loss Adjustment (DLA) mechanism that adaptively balances multiple learning objectives throughout training. By integrating these components, NeuroAlign facilitates a smooth progression from low-level stimulus encoding to high-level semantic alignment, closely reflecting the hierarchical nature of human visual cognition.

### 3.2 VISUAL SALIENCY EXTRACTION

To address Bottleneck 1, namely the feature-physiology mismatch between low-level EEG responses and high-level image features, we design the Visual Saliency Extraction (VSE) module. Under the RSVP paradigm, EEG signals are predominantly associated with low-level visual attributes such as color, brightness, and contrast rather than high-level semantics (Song et al., 2024). Consequently, directly aligning EEG signals with raw RGB images often leads to overfitting on high-frequency, image-specific details (Wu et al., 2025). Inspired by the brains visual system (Creem & Proffitt, 2001), VSE emphasizes physiologically consistent saliency cues by integrating two mechanisms:

rapid saliency detection, which performs a fast pre-scan to simulate attentional shifts, and center-surround antagonism, which enhances fine-grained foregroundbackground separation.

**Rapid Saliency Detection.** This stage extracts salient visual features $S_a$ that align with human attentional mechanisms by combining edge detection and exogenous attention. To simulate the high sensitivity of V1 cortical cells to edges and lines, we first identify the locations of salient regions ("where") by applying a Sobel operator to the grayscale image $I_g$, generating an edge response map $F_e$. Concurrently, to determine the visual attributes to enhance within these regions ("what"), we extract low-level features (luminance $S_l$, contrast $S_d$, and color $S_c$), inspired by the exogenous attention model of (Itti & Koch, 2000). The specific implementations of these features are described in the Appendix A.2.1. A gating mechanism using the sigmoid function $\sigma$ dynamically modulates the contributions of edge feature $E_d$ and the combined exogenous feature $E_x = \sum_{j \in [l,d,c]} \alpha_j S_j$ ($\alpha_j$ represents the weight), enabling adaptive control over the enhancement level. The final saliency feature map $S_a$ is calculated as follows:

$$S_a = [\sigma(\mathcal{N}(E_d)) - \sigma(\mathcal{N}(E_x))] + \eta, \tag{1}$$

where $\mathcal{N}$ represents the normalization operation, and $\eta$ is the constant of the gating mechanism.

**Center-surround Antagonism.** Our approach simulates the center-surround antagonism mechanism observed in the receptive fields of neurons in the retina and visual cortex. We implement this biological principle using a multi-scale Difference of Gaussians (DoG) filter (Marr & Hildreth, 1980). To ensure robustness to variations in feature size, we extend this DoG operation across a spectrum of $N_s$ different image scales (Ghosh et al., 2005), where larger scales capture global structure and smaller scales preserve fine-grained details. The scale-weighted filtered responses are integrated to produce the final antagonism response $R_a$, formulated as follows:

$$R_a = \mathcal{N}\left( \left| \frac{1}{N_s} \sum_{i=1}^{N_s} [\mathcal{G}_c(S_a) - \lambda \mathcal{G}_s(S_a)]_i \right|^\gamma \right) \tag{2}$$

where $|*|^\gamma$ represents the power-law enhancement of the antagonism response, $\mathcal{G}_c$ and $\mathcal{G}_s$ represent the Gaussian filters of the center region and the surrounding region respectively, $\lambda$ is the weight of $\mathcal{G}_s$, $N_s$ denotes the scale number.

The final attention-weighted image $I_a$ is generated by element-wise multiplication of the original image $I$ with the antagonism response map $R_a$:

$$I_a = I \cdot R_a \tag{3}$$

## 3.3 SEMANTIC GUIDANCE ALIGNMENT

While the solutions of Bottleneck 1 focus on mitigating the mismatch between EEG signals and low-level visual features, a second challenge arises from the lack of intra-modality semantic consistency. To address this, we propose the Semantic Guidance Alignment (SGA) loss. Conventional contrastive learning frameworks are susceptible to this issue, as they typically treat all non-paired samples as negatives, potentially dispersing semantically similar samples and failing to enforce intra-class cohesion. The SGA module mitigates this by leveraging semantic priors from a pre-trained vision-language model to create a cross-modality guidance mechanism, which optimizes for semantic consistency in the joint EEG-image feature space.

Specifically, we first employ a large pre-trained language model (BLIP-2 (Li et al., 2023)) to generate textual descriptions of the images. These descriptions, along with the images, are then fed into SLIP's (Mu et al., 2022) encoders to extract image features $\mathbf{V} = \{v_1, \cdots, v_i, \cdots, v_N\}$ and text features $\mathbf{T} = \{t_1, \cdots, t_j, \cdots, t_N\}$, respectively (see Appendix for details). The image-text semantic similarity matrix is constructed as a semantic-guiding prior via softmax normalization:

$$P_{ij} = \begin{cases} \dfrac{\exp(\mathbf{v}_i^\top \mathbf{t}_j / \tau)}{\sum_{n=1}^{N} \exp(\mathbf{v}_i^\top \mathbf{t}_n / \tau)}, & i \neq j \\ 1, & i = j \end{cases} \tag{4}$$

where $\tau$ is an adjustable temperature parameter that controls the concentration of the distribution to provide adaptive semantic guidance strength.

Similarly, an EEG encoder (Song et al., 2025) is used to extract features $\mathbf{E} = \{e_1, \cdots, e_i, \cdots, e_N\}$, from which the EEG-image similarity matrix is constructed as:

$$Q_{ij} = \begin{cases} \dfrac{\exp(\mathbf{e}_i^\top \mathbf{v}_j/\tau)}{\sum_{n=1}^N \exp(\mathbf{e}_i^\top \mathbf{v}_n/\tau)}, & i \neq j \\ 1, & i = j \end{cases} \tag{5}$$

The core objective of SGA is to align EEGimage representations with the semantic prior $P$ derived from imagetext pairs. To this end, we construct the EEGimage similarity distribution $Q$ and minimize its discrepancy from $P$. Specifically, we adopt the KullbackLeibler (KL) divergence as the optimization criterion:

$$\mathcal{L}_{\text{SG}} = D_{\text{KL}}(Q \parallel P) = \sum_{i,j} Q_{ij} \log \frac{Q_{ij}}{P_{ij}} \tag{6}$$

This formulation offers two advantages. First, the asymmetry of the KL divergence allows it to capture the information loss when approximating the semantic prior $P$ with $Q$, thereby preventing noise introduced by reverse transfer. Second, minimizing $D_{\text{KL}}(Q \parallel P)$ can be interpreted as leveraging semantic knowledge from imagetext pairs to guide the EEGimage alignment task. In this way, the EEG feature space is constrained by semantic priors, leading to more consistent and semantically meaningful cross-modality representations.

### 3.4 Dynamic Loss Adjustment

To overcome Bottleneck 3 (the static and unidirectional use of semantic priors), we introduce a Dynamic Loss Adjustment (DLA) strategy that adaptively balances three learning objectives based on their priority: EEG-image alignment loss $\mathcal{L}_{\text{EI}}$ ($\mathcal{L}^1$), image-text alignment loss $\mathcal{L}_{\text{IT}}$ ($\mathcal{L}^2$), and semantic prior guidance loss $\mathcal{L}_{\text{SG}}$ ($\mathcal{L}^3$). The overall loss is formulated as

$$\mathcal{L} = w^1 \mathcal{L}^1 + w^2 \mathcal{L}^2 + w^3 \mathcal{L}^3 \tag{7}$$

Instead of fixing the weight of each part, DLA introduces a dynamic strategy inspired by the brain's hierarchical visual processing, which evolves from initial stimulus-driven encoding to subsequent semantically-guided integration. To mimic this cognitive progression, DLA dynamically modulates the weights $\mathbf{w}_t^k$ based on the learning state of each objective, quantified by the gradient magnitude $g_t^k$ and its relative velocity $r_t^k$. This mechanism ensures that the optimization focus shifts organically over time. The weights are calculated as:

$$\mathbf{w}_t^k = \text{softmax}\left(g_t^k \cdot (1 + r_t^k)/T\right), \tag{8}$$

where

$$g_t^k = \left\| \frac{\partial \mathcal{L}_t^k}{\partial \boldsymbol{\theta}_k} \right\|_2, \quad r_t^k = \frac{|g_t^k - g_{t-1}^k|}{g_{t-1}^k + \epsilon} \tag{9}$$

Here, $T$ is a temperature parameter controlling the smoothness of the contribution, $\theta$ represents learnable parameters, and $\epsilon$ is a small constant to prevent division by zero. This design ensures a fluid integration of the EEG-image, image-text, and semantic guidance objectives, aligning the learning process with both cognitive principles and data-driven dynamics. Consequently, the model's learning process evolves, demonstrating a shift from low-level, stimulus-driven analysis toward high-level, semantically guided synthesis with the evolution of the training process.

## 4 Experiments

To systematically evaluate the performance of the proposed NeuroAlign framework, we conducted comprehensive experiments on two publicly available datasets. Primary evaluation and ablation studies were performed on the THINGS-EEG2 dataset (Gifford et al., 2022), with additional cross-modality validation carried out using the THINGS-MEG dataset (Hebart et al., 2023). This section describes the datasets and experimental setup, followed by a detailed analysis of the results. Details regarding datasets, preprocessing procedures, hyperparameter configurations, hardware specifications, ablation study, and retrieval case analysis are provided in the Appendix A.3.

### 4.1 RESULTS

We evaluate the proposed NeuroAlign framework on the THINGS-EEG2 and THINGS-MEG datasets under a challenging 200-way zero-shot image retrieval setup. The evaluation follows two rigorous paradigms: (1) **Intra-subject evaluation**, in which models are trained and tested on data from the same individual; and (2) **Inter-subject evaluation**, where a model is trained on data from all but one subject and tested on the held-out individual, thereby critically assessing inter-subject generalization capability.

Table 1: Top-1 and Top-5 accuracy (%) for 200-way zero-shot retrieval on THINGS-EEG2

| Method | Subject 1 | | Subject 2 | | Subject 3 | | Subject 4 | | Subject 5 | | Subject 6 | | Subject 7 | | Subject 8 | | Subject 9 | | Subject 10 | | Avg | |
|---|---|---|---|---|---|---|---|---|---|---|---|---|---|---|---|---|---|---|---|---|---|---|
| | top-1 | top-5 | top-1 | top-5 | top-1 | top-5 | top-1 | top-5 | top-1 | top-5 | top-1 | top-5 | top-1 | top-5 | top-1 | top-5 | top-1 | top-5 | top-1 | top-5 | top-1 | top-5 |
| **Intra-subject**: train and test on one subject | | | | | | | | | | | | | | | | | | | | | | |
| BraVL | 6.1 | 17.9 | 4.9 | 14.9 | 5.6 | 17.4 | 5.0 | 15.1 | 4.0 | 13.4 | 6.0 | 18.2 | 6.5 | 20.4 | 8.8 | 23.7 | 4.3 | 14.0 | 7.0 | 19.7 | 5.8 | 17.5 |
| NICE | 13.2 | 39.5 | 13.5 | 40.3 | 14.5 | 42.7 | 20.6 | 52.7 | 10.1 | 31.5 | 16.5 | 44.0 | 17.0 | 42.1 | 22.9 | 56.1 | 15.4 | 41.6 | 17.4 | 45.8 | 16.1 | 43.6 |
| NICE-SA | 13.3 | 40.2 | 12.1 | 36.1 | 15.3 | 39.6 | 15.9 | 49.0 | 9.8 | 34.4 | 14.2 | 42.4 | 17.9 | 43.6 | 18.2 | 50.2 | 14.4 | 38.7 | 16.0 | 42.8 | 14.7 | 41.7 |
| NICE-GA | 15.2 | 40.1 | 13.9 | 40.1 | 14.7 | 42.7 | 17.6 | 48.9 | 9.0 | 29.7 | 16.4 | 44.4 | 14.9 | 43.1 | 20.3 | 52.1 | 14.1 | 39.7 | 19.6 | 46.7 | 15.6 | 42.8 |
| NICE-LLM | 15.0 | 48.5 | 17.5 | 46.0 | 19.5 | 51.5 | 29.0 | 60.0 | 13.5 | 44.5 | 18.0 | 55.5 | 22.0 | 55.5 | 36.5 | 55.0 | 20.0 | 68.5 | 22.0 | 52.0 | 21.3 | 53.4 |
| ATM-S | 25.6 | 60.4 | 22.0 | 54.5 | 25.0 | 62.4 | 31.4 | 60.9 | 12.9 | 43.0 | 21.3 | 51.1 | 30.5 | 61.5 | 38.8 | 72.0 | 34.4 | 51.5 | 29.1 | 63.5 | 28.5 | 60.4 |
| UBP | 33.0 | 70.4 | 45.5 | 73.5 | 43.5 | 78.0 | 44.5 | 79.5 | 36.5 | 67.5 | 51.0 | 79.0 | 41.0 | 75.0 | 58.0 | 82.0 | 44.5 | 74.5 | 59.0 | 86.0 | 45.7 | 76.6 |
| **Ours** | 42.0 | 74.5 | 39.5 | 72.0 | 51.5 | 80.0 | 51.0 | 81.5 | 38.0 | 68.0 | 56.5 | 83.0 | 46.5 | 73.5 | 60.5 | 85.0 | 43.0 | 78.0 | 52.0 | 84.0 | **48.1** | **78.1** |
| **Inter-subject**: leave one subject out for test | | | | | | | | | | | | | | | | | | | | | | |
| BraVL | 2.3 | 8.0 | 1.5 | 6.3 | 1.4 | 5.9 | 1.7 | 6.7 | 1.5 | 5.6 | 1.8 | 7.2 | 2.1 | 8.1 | 2.2 | 7.6 | 1.6 | 6.4 | 2.3 | 8.5 | 1.8 | 7.0 |
| NICE | 7.6 | 22.8 | 5.9 | 20.5 | 6.0 | 22.3 | 6.3 | 20.7 | 4.4 | 18.3 | 5.6 | 22.2 | 5.6 | 19.7 | 6.3 | 22.0 | 5.7 | 17.6 | 8.4 | 28.3 | 6.2 | 21.4 |
| NICE-SA | 7.0 | 22.6 | 6.6 | 23.2 | 7.5 | 23.7 | 5.4 | 21.4 | 6.4 | 22.2 | 7.5 | 22.5 | 3.8 | 19.1 | 8.5 | 24.4 | 7.4 | 22.3 | 9.8 | 29.6 | 7.0 | 23.1 |
| NICE-GA | 5.9 | 21.4 | 6.4 | 22.7 | 5.5 | 20.1 | 6.1 | 21.0 | 4.7 | 19.5 | 6.2 | 22.5 | 5.9 | 19.1 | 7.3 | 25.3 | 4.8 | 18.3 | 6.2 | 26.3 | 5.9 | 21.6 |
| NICE-LLM | 7.5 | 28.0 | 9.5 | 30.0 | 10.5 | 27.5 | 10.0 | 31.5 | 5.0 | 21.0 | 11.4 | 29.5 | 7.5 | 22.5 | 6.0 | 25.5 | 7.5 | 23.0 | 6.2 | 26.3 | 9.2 | 27.7 |
| ATM-S | 10.5 | 26.8 | 7.1 | 24.8 | 11.9 | 33.8 | 14.7 | 39.4 | 7.0 | 23.9 | 11.1 | 35.8 | 16.1 | 43.5 | 15.0 | 40.3 | 4.9 | 22.7 | 20.5 | 46.5 | 11.8 | 33.7 |
| UBP | 12.0 | 33.5 | 12.0 | 37.5 | 10.0 | 24.5 | 13.2 | 31.5 | 10.5 | 29.6 | 13.5 | 30.5 | 10.4 | 26.0 | 9.0 | 33.5 | 9.5 | 31.5 | 18.5 | 43.2 | 11.9 | 32.1 |
| **Ours** | 14.5 | 44.5 | 19.0 | 42.0 | 10.0 | 28.5 | 15.5 | 37.5 | 12.0 | 27.5 | 18.0 | 39.5 | 11.0 | 34.0 | 14.0 | 32.5 | 13.0 | 32.0 | 18.0 | 46.0 | **14.5** | **36.4** |

**Results on the THINGS-EEG2 dataset (Table 1).** In the **intra-subject** evaluation, NeuroAlign achieved average Top-1 and Top-5 accuracies of **48.1%** and **78.1%**, respectively, in the 200-way zero-shot image retrieval task, significantly outperforming all baseline models. Under the more challenging **inter-subject** evaluation, the model still attained a Top-1 accuracy of **14.5%** and a Top-5 accuracy of **36.4%**, demonstrating robust inter-subject generalization capability. Specifically, NeuroAlign achieved the best Top-1 performance on 8 out of 10 subjects, with particularly outstanding results on Subjects 6, 8, and 10 (Top-1 accuracies of $56.5\%, 60.5\%, and 52.0\%$, respectively). Compared to the current state-of-the-art model UBP, NeuroAlign improved the average Top-1 accuracy by 2.4 percentage points while delivering leading or comparable Top-5 performance across all subjects, underscoring its stable retrieval capability under diverse neural response characteristics.

Table 2: Top-1 and Top-5 accuracy (%) for 200-way zero-shot retrieval on THINGS-MEG.

| Method | Subject 1 | | Subject 2 | | Subject 3 | | Subject 4 | | Avg | |
|---|---|---|---|---|---|---|---|---|---|---|
| | top-1 | top-5 | top-1 | top-5 | top-1 | top-5 | top-1 | top-5 | top-1 | top-5 |
| **Intra-subject**: train and test on one subject | | | | | | | | | | |
| NICE | 9.6 | 27.8 | 18.5 | 47.8 | 14.2 | 41.6 | 9.0 | 26.6 | 12.8 | 36.0 |
| NICE-SA | 9.8 | 27.8 | 18.6 | 46.4 | 10.5 | 38.4 | 11.7 | 27.2 | 12.7 | 35.0 |
| NICE-GA | 8.7 | 30.5 | 21.8 | 56.6 | 16.5 | 49.7 | 10.3 | 32.3 | 14.3 | 42.3 |
| NICE-LLM | 9.0 | 32.5 | 19.5 | 50.0 | 17.5 | 48.0 | 12.5 | 31.5 | 14.6 | 40.5 |
| UBP | 15.5 | 40.5 | 45.0 | 77.5 | 27.0 | 57.0 | 13.5 | 37.5 | 25.3 | 53.1 |
| **Ours** | 17.0 | 40.5 | 42.0 | 75.5 | 31.0 | 61.5 | 16.0 | 35.5 | **26.5** | **53.3** |
| **Inter-subject**: leave one subject out for test | | | | | | | | | | |
| NICE-LLM | 3.0 | 9.5 | 1.0 | 9.5 | 3.0 | 7.5 | 3.0 | 9.0 | 2.5 | 8.9 |
| UBP | 1.5 | 4.5 | 3.5 | 14.5 | 1.0 | 8.1 | 2.0 | 9.5 | 2.0 | 9.2 |
| **Ours** | 4.5 | 11.0 | 5.5 | 16.0 | 2.5 | 15.5 | 3.0 | 10.0 | **3.9** | **13.1** |

**Results on the THINGS-MEG dataset (Table 2)**. Under the Intra-subject setting, NeuroAlign achieved average Top-1 and Top-5 accuracies of **26.5%** and **53.3%**, respectively, outperforming all baseline models. In the more challenging Inter-subject setting, the model attained a Top-1 accuracy of **3.9%** and Top-5 accuracy of **13.1%**, demonstrating significant improvements over existing methods. Notably, NeuroAlign achieved the best or competitive performance across all four subjects in both evaluation paradigms. For intra-subject retrieval, it showed particularly strong performance on Subject 2 (Top-1: **42.0%**) and Subject 3 (Top-1: **31.0%**), while maintaining robust results on the other subjects. The consistent superiority in inter-subject evaluation highlights the model's effective generalization capability across different MEG recording sessions and individual neural response patterns.

## 4.2 ANALYSIS

### 4.2.1 VISUAL SALIENCY ANALYSIS

Figure 3 illustrates the effectiveness of the VSE module in capturing visually and physiologically relevant features. We evaluated retrieval accuracy using two types of temporal windows: forwardincreasing windows $[0, t]$ from stimulus onset to time $t$, and backwarddecreasing windows $[t, 1000]$ from $t$ to 1000 ms. As shown in Figure 3 (a) and Figure 3 (b), models incorporating the VSE module exhibit faster accuracy growth and significantly higher retrieval performance in early time windows (100-500 ms) compared to ablated versions, indicating that VSE-derived features align well with early visual processing stages. Furthermore, results in Figure 3 (c) shows that the model can retrieve images with similar color distributions, object contours, and overall compositions. This indicates that the VSE module is able to extract low-level visual features like color, edges, and shapes for similarity matching. The analysis of the EEG channels is provided in Appendix A.3.5.

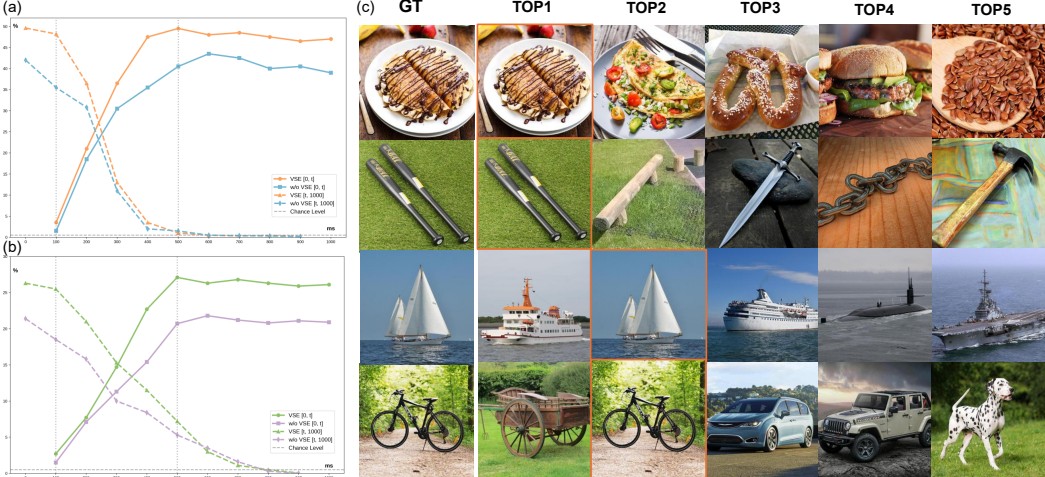

Figure 3: Comparison of Time Accuracy and Ground Truth. (a) Top1 retrieval accuracy under different time windows for THINGS-EEG2. Integration of the VSE module yields significant gains in the 100-500 ms window, reflecting better alignment with early visual responses. (b) Corresponding results for THINGS-MEG. (c) Top5 retrieval examples on THINGS-EEG2. The model captures color, edge, and structural cues to achieve consistent category-level semantic distinctions.

### 4.2.2 SEMANTIC CONSISTENCY ANALYSIS

To quantitatively evaluate the impact of semantic guidance on brain-vision alignment, we conducted Representation Similarity Analysis (Cichy & Oliva, 2020) following established protocols (Song et al., 2024). We selected samples from the four most frequent core categories in the THINGS dataset (Hebart et al., 2019) and visualized their EEG-image feature similarity distributions using t-SNE dimensionality reduction (Maaten & Hinton, 2008).

As shown in Figure 4, the feature distribution before EEG-image alignment exhibits substantial category overlap with minimal semantic structure (Figure 4 (a). Crucially, when ablating the SGA module, the categories become more separable but still exhibit notable overlap (Figure 4 (b). In contrast, NeuroAlign with complete semantic guidance produces tightly clustered, semantically coherent representations where samples from the same category converge while maintaining clear inter-class boundaries (Figure 4 (c). This demonstrates that SGA module constraints are essential for enforcing intra-class consistency, enabling the model to transcend mere visual similarity and capture genuine semantic relationships across modalities.

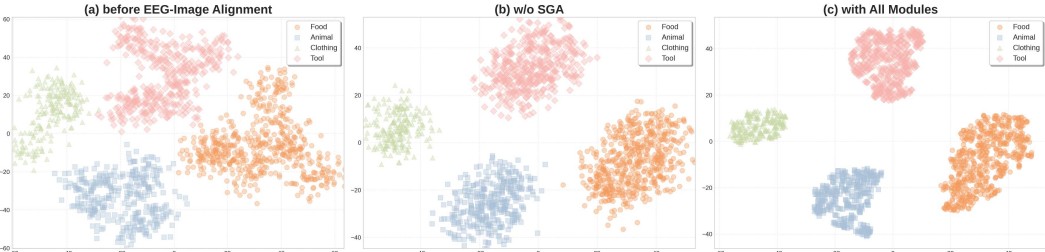

Figure 4: Semantic consistency analysis demonstrating SGA's advantage in intra-class coherence. (a) shows the feature distribution before EEG-image alignment, where categories are largely overlapping and semantically indistinct; (b) shows the distribution without semantic guidance alignment, categories remain poorly separated despite improved visual feature alignment; (c) demonstrates that NeuroAlign produces well-separated and semantically coherent clusters, confirming that the model learns semantically consistent cross-modality representations.

### 4.2.3 DYNAMIC LOSS ADJUSTMENT ANALYSIS

**Quantitative Evaluation.** As shown in Figure 5(a), NeuroAlign significantly outperforms the NICE-LLM baseline across all Top-k metrics. To quantitatively evaluate the contribution of the Dynamic Loss Adjustment (DLA) mechanism, we applied it as a standalone optimization module to the NICE-LLM architecture. Although NICE-LLM already leverages the language modality for alignment, the introduction of DLA yielded substantial performance gains, boosting its Top-1 accuracy from 21.3% to 28.9% and its Top-5 accuracy from 53.4% to 61.8%. Furthermore, our ablation study on NeuroAlign (see Section 4.2.4) confirms that removing the DLA module leads to a performance drop, underscoring its critical role in adaptively balancing the alignment between modalities.

**Robustness Analysis.** The weight dynamics in Figure 5(b) simulate a cognitive progression from stimulus-driven to task-driven processing. We validated the robustness of DLA against subject and random seed variations through two rigorous experiments. First, for a single subject (Subject 8), repeated training runs using five different random seeds exhibit minimal variance (std < 0.1), as indicated by the shaded area. Second, the evolution curve is highly consistent across all 10 subjects (details in Appendix A.2.3). This deterministic convergence suggests that DLA captures generalizable principles of brain-image-text learning, rather than overfitting to specific data distributions or individual differences.

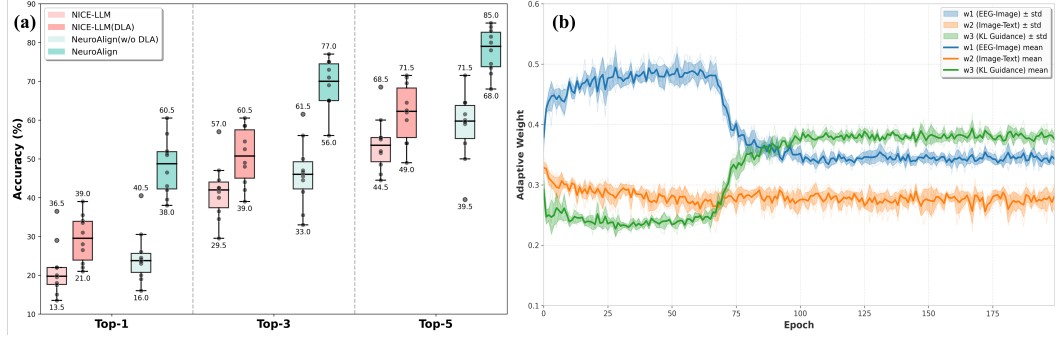

Figure 5: Effectiveness and Robustness of the DLA Mechanism. (a) Quantitative Evaluation: Integrating DLA into both NeuroAlign and the NICE-LLM baseline significantly improves Top-k retrieval accuracy. (b) Robustness Analysis: The weight evolution curves demonstrate a robust transition from a stimulus-driven to a task-driven process. For Subject 8, repeated training runs using five different random seeds exhibit minimal variance (std < 0.1). This indicates that DLA captures the underlying principles of brain-image-text cross-modal learning independent of initialization noise.

### 4.2.4 ABLATION STUDY

We conducted systematic ablation studies on the THINGS-EEG2 and THINGS-MEG datasets to isolate and quantify the contribution of each component. By comparing the full NeuroAlign framework against variants lacking specific components, we evaluated the necessity of three core modules: Visual Saliency Extraction (VSE), Semantic Guidance Alignment (SGA), and Dynamic Loss Adjustment (DLA).

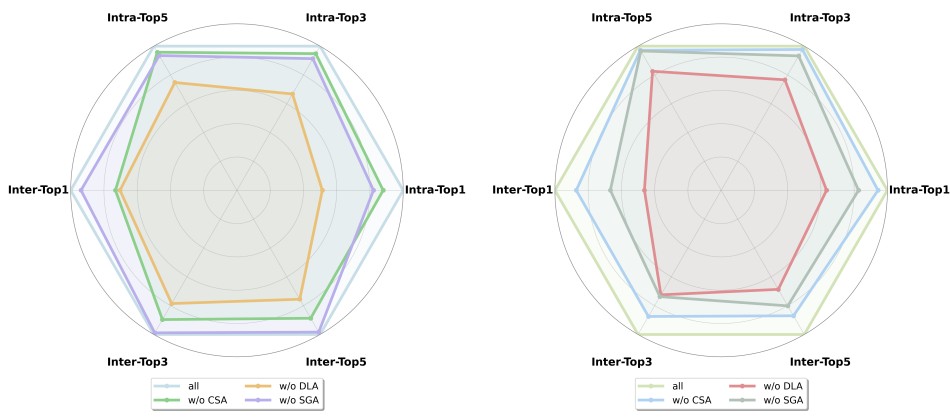

Figure 6: Ablation results of NeuroAlign on EEG/MEG-based image retrieval. Left: Intra-subject and inter-subject classification accuracy on the THINGS-EEG2 dataset after removing VSE, SGA, and DLA. Right: Corresponding results on the THINGS-MEG dataset. Removing any single component leads to performance degradation, with the absence of DLA causing the largest drops, confirming the necessity of all modules for robust cross-modality alignment.

As shown in Figure 6, the removal of any individual component consistently reduced performance under both intra-subject and inter-subject evaluation settings. The most significant declines occurred when DLA was ablated. Specifically, in the THINGS-EEG2 inter-subject setting, removing DLA led to a sharp decline of 23.4% in Top-1 accuracy and a decrease of 19.7% in Top-5 accuracy. This illustrates that DLA provides a critical dynamic integration that maximizes the effectiveness of VSE and SGA. These results confirm the necessity of each module and demonstrate that NeuroAlign's effectiveness arises from their joint integration, reflecting the hierarchical processing and adaptive weighting mechanisms characteristic of human visual cognition. Table 4 provides a detailed quantitative comparison.

## 5 CONCLUSION

This paper presents NeuroAlign, a neuroscience-inspired framework for brainvisual cross-modality alignment that dynamically guides the integration of brain signals and visual representations by leveraging language as a supervisory modality. The framework effectively addresses three key limitations in existing methods: feature-physiology mismatch, lack of intra-modality semantic consistency, and static multi-objective optimization. Extensive experiments show that NeuroAlign outperforms previous SO approaches, achieving Top-1/Top-5 accuracies of $48.1\%/78.1\%$ in intra-subject evaluation and $14.5\%/36.4\%$ in inter-subject evaluation on the THINGS-EEG2 dataset. Comparable improvements are observed on the THINGS-MEG dataset, confirming the generalizability of the approach. While NeuroAlign represents a meaningful step toward modeling brainvision correspondence, it remains a preliminary computational probe into complex neural processes. The specific neural mechanisms captured by the models alignment process warrant further investigation. Future work should extend our framework to image reconstruction, incorporate richer neurobiological constraints and explore representational hierarchies to better simulate and interpret the brain's visual processing pathways.

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

## A  APPENDIX

### A.1  LARGE LANGUAGE MODELS STATEMENT

LLM played a significant role in polishing this paper and retrieving related work.

### A.2  METHODOLOGICAL DETAILS

The overall pipeline of NeuroAlign is summarized in Algorithm 1, which comprises four core processing stages: Visual Saliency Extraction (VSE), Tri-Modality Feature Extraction, Semantic Guidance Alignment, and Dynamic Loss Adjustment.

---

**Algorithm 1** NeuroAlign: A Neuroscience-Inspired Framework for Visual EEG Decoding

---

1: **Input:** Paired image-EEG batch $\{V_i, E_i\}_{i=1}^N$     ▷ Assume $E_i$ is evoked by visual stimulus $V_i$
2: **Models:** EEG Encoder $F_e$, Visual Encoder $F_v$ (SLIP), Text Encoder $F_t$ (SLIP)
3: **Parameters:** Loss weights $\mathbf{w}_t = [w_t^1, w_t^2, w_t^3]$, temperature $\tau$, step $t$
    ▷ *Stage 1: Visual Saliency Extraction (VSE)*
4: **for** each image $V_i$ **do**
5:     Extract edge map $E_d$ and low-level features $\{S_l, S_d, S_c\}$ from $V_i$
6:     $S_a \leftarrow \sigma(\mathcal{N}(E_e)) - \sigma(\mathcal{N}(\sum \alpha_j S_j)) + \gamma$     ▷ Fuse edge and exogenous features
7:     Apply multi-scale DoG filter to $S_a$ to obtain $R_a$     ▷ Center-surround antagonism
8:     $I_a \leftarrow V_i \cdot R_a$     ▷ Generate saliency-enhanced image
9: **end for**
    ▷ *Stage 2: Tri-Modality Feature Extraction (TFE)*
10: $\{T_i\}_{i=1}^N \leftarrow$ BLIP-2($\{V_i\}_{i=1}^N$)     ▷ Generate textual descriptions
11: $e_i \leftarrow \text{Norm}(P_e(F_e(E_i)))$     ▷ EEG feature
12: $v_i \leftarrow \text{Norm}(P_v(F_v(I_a)))$     ▷ Visual feature
13: $t_i \leftarrow \text{Norm}(P_t(F_t(T_i)))$     ▷ Text feature
    ▷ *Stage 3: Semantic Guidance Alignment (SGA)*
14: $P_{ij} \leftarrow \frac{\exp(v_i \cdot t_j / \tau)}{\sum_n \exp(v_i \cdot t_n / \tau)}$     ▷ Teacher distribution (image-text)
15: $Q_{ij} \leftarrow \frac{\exp(e_i \cdot v_j / \tau)}{\sum_n \exp(e_i \cdot v_n / \tau)}$     ▷ Student distribution (EEG-image)
16: $\mathcal{L}^3 \leftarrow \mathcal{L}_{\text{SG}} = \frac{1}{N} \sum_i D_{\text{KL}}(Q_i \parallel P_i)$     ▷ KL divergence loss
    ▷ *Stage 4: Dynamic Loss Adjustment (DLA)*
17: $\mathcal{L}^1 \leftarrow \text{InfoNCE}(\{e_i\}, \{v_i\}, \tau)$     ▷ EEG-Image contrastive loss
18: $\mathcal{L}^2 \leftarrow \text{InfoNCE}(\{v_i\}, \{t_i\}, \tau)$     ▷ Image-Text contrastive loss
19: Compute $g_t^k$ and $r_t^k$ for $k \in \{1, 2, 3\}$ based on Eq. (9)     ▷ Training dynamics
20: Update weights: $w_t^k \leftarrow \text{softmax}\left(\frac{g_t^k \cdot (1+r_t^k)}{T}\right)$     ▷ Adaptive weighting (Eq. 8)
21: $\mathcal{L}_{\text{total}} = w_t^1 \mathcal{L}^1 + w_t^2 \mathcal{L}^2 + w_t^3 \mathcal{L}^3$
22: Update all parameters via $\nabla \mathcal{L}_{\text{total}}$

---

### A.2.1 VISUAL SALIENCY EXTRACTION (VSE)

This section provides supplementary mathematical formulations for the Visual Saliency Extraction module (Section 3.2). The VSE module consists of two main components: Rapid Saliency Detection and Center-Surround Antagonism. The module extracts and combines four key visual features (edge, luminance, contrast, and color) to generate EEG-aligned saliency maps.

The rapid saliency detection stage begins by converting the input RGB image $V_i$ to grayscale $I_g$. We apply Sobel operators for edge detection:

$$F_e = \sqrt{(\nabla_x I_g)^2 + (\nabla_y I_g)^2}, \tag{10}$$

where $\nabla_x$ and $\nabla_y$ are horizontal and vertical Sobel operators, respectively.

After that, we extract low-level features (luminance $S_l$, contrast $S_d$, and color $S_c$):

$$S_l = |I_g - \mu|, \quad S_d = |I_g - \sigma_l|, \quad S_c = \sum_{i \neq j} |C_i - C_j| \tag{11}$$

where $\mu$ and $\sigma_l$ are the global mean and standard deviation of $I_g$ respectively, and $C_i, C_j$ represent different color channels.

The center-surround antagonism stage enhances foreground-background separation using multi-scale Difference of Gaussians (DoG) filtering. For each scale $s \in \{1, 2\}$, we define center and surround Gaussian filters:

$$\mathcal{G}_c(x, y, s) = \frac{1}{2\pi\sigma_c^2} \exp\left(-\frac{x^2 + y^2}{2\sigma_c^2}\right) \tag{12}$$

$$\mathcal{G}_s(x, y, s) = \frac{1}{2\pi\sigma_s^2} \exp\left(-\frac{x^2 + y^2}{2\sigma_s^2}\right) \tag{13}$$

where $\mathcal{G}_c$ and $\mathcal{G}_s$ are the center and surround Gaussian functions, respectively. The terms $x$ and $y$ are pixel coordinates. $\sigma_c$ and $\sigma_s$ are their standard deviations, with $\sigma_c = scale$ and $\sigma_s = 2 \times scale$ defining the broader surround area relative to the center.

This comprehensive pipeline ensures that the VSE module emphasizes visual characteristics that align with EEG responses while suppressing irrelevant high-frequency details, thereby improving the cross-modal alignment between brain signals and visual stimuli in the RSVP paradigm.

To illustrate the functionality of each process, we provide a concrete example in Figure 7. First, the original image (Fig.7(d)) is processed by the Rapid Saliency Detection (RSD) module to generate a grayscale map emphasizing edges and contours (Fig.7(a)). This step clearly delineates the structure of the foreground object (e.g., the antelope) while transforming the background into structureless textures. Subsequently, the Center-Surround Antagonism (CSA) module operates on this basis to enhance the contrast between salient regions and the background. The resulting output is the pixel-wise attention map, $R_a$ (Fig.7(b)). To visualize the attention distribution more intuitively, Fig.7(c) presents the heatmap corresponding to $R_a$ . Finally, by performing element-wise multiplication between $R_a$ and the original image, we obtain the final weighted image, $I_a$ (Fig.7(e)), with its corresponding heatmap distribution shown in Fig.7(f). Comparing Fig.7(d) with Fig.7(e), it is evident that the VSE process effectively suppresses task-irrelevant background noise while preserving and enhancing the salient features of the visual subject, thereby providing a more focused visual input for the subsequent alignment between brain signals and images.

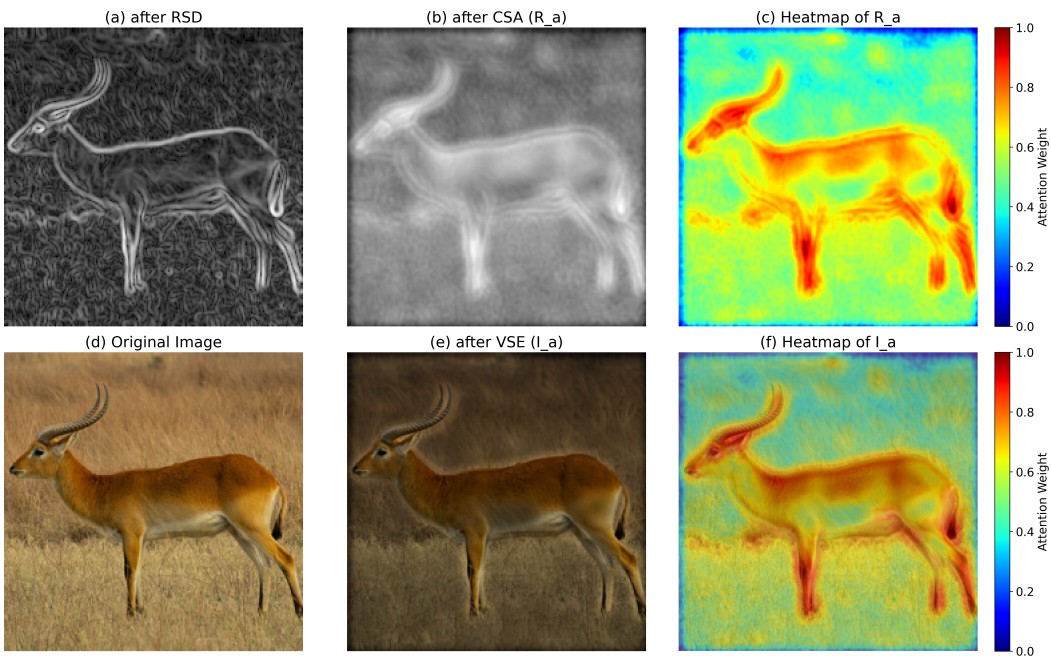

Figure 7: (a) Output after the Rapid Saliency Detection (RSD) module, highlighting edge structures. (b) The pixel-wise attention map $R_a$ generated by the Center-Surround Antagonism (CSA) module. (c) Heatmap visualization of $R_a$. (d) The original input image. (e) The final attention-weighted image $I_a$ after VSE, showing suppressed background and highlighted foreground. (f) Heatmap visualization of the final output $I_a$.

### A.2.2 TRI-MODALITY FEATURE EXTRACTION (TFE)

To learn unified and semantically aligned representations across neural, visual, and linguistic modalities, we adopt three specialized encoders that extract features from EEG signals, images, and text, respectively. Raw inputs from each modality are first encoded into high-dimensional features via dedicated encoders, which are then projected into a shared embedding space $\mathbb{R}^d$.

**EEG Encoder.** To capture spatio-temporal patterns in EEG signals, we employ a Temporal-Spatial Convolutional Network (TSConv) encoder, a design widely validated in EEG decoding literature (Song et al., 2025). As implemented in our model, this process begins with a temporal convolution to extract time-series features, followed by an average pooling layer for downsampling. Subsequently, a spatial convolution aggregates information across EEG channels. The resulting feature map is then linearly projected and rearranged into a sequence of embedding vectors. These vectors are flattened into a single high-dimensional representation, which is finally mapped by projection $P_i$ to produce the final EEG embedding $E_i$ within the shared multi-modal space.

**Visual Encoder.** We employ SLIP (Mu et al., 2022) as our visual encoder, which incorporates both cross-modal alignment and intra-modal self-supervision. Given an input image, we first utilize the Visual Saliency Extraction (VSE) module to enhance its salient regions, and then encode these enhanced regions through the image encoder and the corresponding projection network $P_i$.

**Text Encoder.** For the textual modality, we first use BLIP-2 to generate descriptive captions for each image to provide high-quality semantic supervision. These generated captions are then processed by the text encoder from a pre-trained SLIP model.

### A.2.3 DYNAMIC LOSS ADJUSTMENT (DLA)

To validate the stability and generalizability of our DLA mechanism, we monitored the evolution of the adaptive weights $(w_1, w_2, w_3)$ across 10 subjects from the THINGS-EEG2 dataset. As illustrated in Figure 8, during the early training stages, the model prioritizes direct image-EEG alignment ($w_1$ dominates), establishing a coarse visual grounding. As training evolves, $w_1$ gradually decays while $w_3$ steadily increases, shifting the focus towards high-level semantic alignment. The system finally converges to a stable state, balancing the contributions of both pathways. Despite the inter-subject variability inherent in EEG signals, the weight evolution trajectories demonstrate remarkable consistency across all individuals.

### A.3 EXPERIMENTAL DETAILS

### A.3.1 DATASETS

**THINGS-EEG2 Dataset:** This dataset contains EEG recordings from 10 subjects under a rapid serial visual presentation (RSVP) paradigm. Visual stimuli consisted of natural images spanning 1,854 unique semantic categories. The dataset is partitioned into a training set containing 1,654 categories (with 10 images per category) and a disjoint test set of 200 categories (with 1 image per category). To ensure robust signal acquisition, each training image was presented 4 times, while each test image was repeated 80 times.

**THINGS-MEG Dataset:** This dataset contains MEG recordings from four subjects using 271 channels during image viewing tasks. The training set includes 1,654 concepts, each represented by 12 distinct images, with each image shown once. The test set consists of 200 concepts, each represented by a single image, repeated 12 times. During data collection, each stimulus was presented for 500 ms, followed by a blank screen of variable duration.

### A.3.2 DATA PREPROCESSING

For **THINGS-EEG2**, preprocessing began with event detection and channel selection on the raw EEG data. Time windows from 0.2 s to 1.0 s relative to stimulus onset were extracted. Baseline correction was applied using the 200 ms pre-stimulus interval, and signals from 63 electrodes were retained and downsampled to 250 Hz. Following the procedure described in (Song et al., 2024), we averaged repeated EEG trials corresponding to the same image to improve the signal-to-noise ratio.

For **THINGS-MEG**, we adopted the preprocessing pipeline of (Song et al., 2024): data were segmented into trials spanning 0-1000 ms after stimulus onset, bandpass-filtered between 0.1 and 100 Hz, downsampled to 200 Hz, and baseline-corrected. To enhance signal quality, all MEG repetitions for each image were averaged.

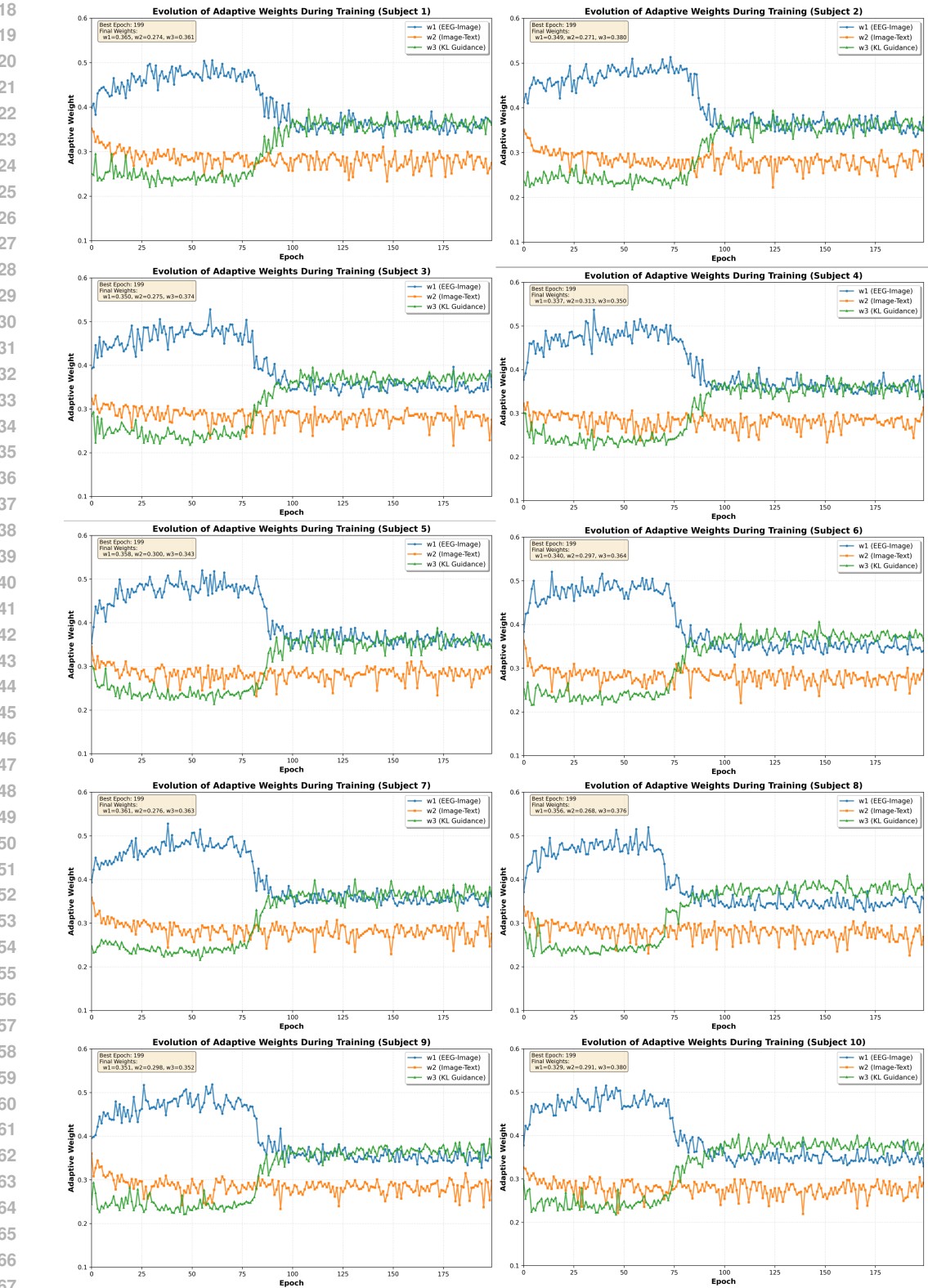

Figure 8: **Consistency of DLA Weight Evolution Across Different Subjects.** This figure illustrates the evolution of the adaptive weights $w_1$, $w_2$, and $w_3$ for 10 different subjects on the THINGS-EEG2 dataset during training. Despite individual variations in the data, the weight trajectories for all subjects exhibit a highly similar pattern: $w_1$ (image-EEG alignment) initially dominates and then declines, while $w_3$ (semantic guidance) gradually increases before stabilizing.

### A.3.3 EVALUATION PROTOCOLS

We employ two complementary evaluation protocols on the THINGS-EEG2 and THINGS-MEG datasets to provide a comprehensive assessment of the models performance and generalization capability:

**Intra-Subject Evaluation:** We follow the original dataset partitioning. The officially provided test set is used directly. From the original training set, we randomly shuffle the samples and reserve the first 740 instances for validation, with the remainder used for training. This setup ensures a standardized and reproducible framework for within-subject analysis.

**Inter-Subject Evaluation:** This protocol evaluates the model's ability to generalize across individuals. The test data from a target subject are held out exclusively for testing. Training data from all other subjects (excluding the target subject) are combined to form a pooled training set. From this pool, we randomly shuffle the samples and allocate the first $740 \times 9$ examples to the validation set, with the remaining samples used for training. This approach strictly prevents data leakage across subjects and adheres to leave-one-subject-out validation standards.

### A.3.4 IMPLEMENTATION DETAILS

Our framework is implemented in PyTorch and trained on a NVIDIA A100 GPU. For each experimental run, we randomly sample 740 trials from the training data to construct a validation set. Models are trained for 200 epochs, with the best checkpoint selected according to the lowest validation loss. Final evaluation is conducted in a single pass over the test set upon completion of training.

Due to the pre-extraction of image features, the training process for each subject requires approximately 10 minutes with a batch size of 1000. Key architectural parameters in the TSConv module include $k = 40$, $m_1 = 25$, $m_2 = 51$, and $s = 5$, which are determined through preliminary experiments. We employ the Adam optimizer with its default momentum parameters. Comprehensive hyperparameter settings are detailed in Table 3.

Table 3: Summary of Hyperparameter Settings

| Category | Symbol | Description | Value/Setting |
|---|---|---|---|
| **Data Pre-processing** | | | |
| Batch Size | $B$ | Training batch size | 256 |
| Image Size | - | Resize and CenterCrop dimensions | $224 \times 224$ |
| **Visual Saliency Extraction (VSE)** | | | |
| DoG Scale Set | $scale$ | Multi-scale set for center-surround antagonism | [1, 2, 4] |
| Center Gaussian $\sigma$ | $\sigma_c$ | DoG center Gaussian kernel std | $1.0 \times$ scale |
| Surround Gaussian $\sigma$ | $\sigma_s$ | DoG surround Gaussian kernel std | $2.0 \times$ scale |
| Sobel Kernel Size | $k_{sobel}$ | Edge detection kernel size | 7 |
| Antagonism exponent | $\gamma$ | power exponent for antagonism | 1.2 |
| Surround Gaussian weight | $\lambda$ | Gaussian region intensity | 0.5 |
| **Model Architecture** | | | |
| EEG Projection Input Dim | $d_{eeg\_in}$ | Input dimension for the EEG projection head | 1440 |
| Image Projection Input Dim | $d_{img\_in}$ | Input dimension for the image projection head | 512 |
| Text Projection Input Dim | $d_{txt\_in}$ | Input dimension for the text projection head | 512 |
| Shared Projection Dim | $d_{proj}$ | Output dimension for all projection heads | 256 |
| Projection Dropout Rate | $p_{drop}$ | Dropout rate for the projection layers | 0.2 |
| **Training Configuration** | | | |
| Training Epochs(EEG) | $E_1$ | Total training epochs | 200 |
| Training Epochs(MEG) | $E_2$ | Total training epochs | 100 |
| Learning Rate | $r$ | Initial learning rate | 0.0002 |
| Optimizer | - | Optimization algorithm | AdamW |
| $\beta_1$ | $\beta_1$ | AdamW first momentum coefficient | 0.5 |
| $\beta_2$ | $\beta_2$ | AdamW second momentum coefficient | 0.999 |
| Weight Decay (EEG) | $\lambda_{wd\_eeg}$ | Weight decay for EEG parameters | 0.001 |
| Weight Decay (Image) | $\lambda_{wd\_img}$ | Weight decay for image parameters | 0.005 |
| Validation Set Size | $N_{val}$ | Number of validation samples | 740 |
| **Loss Function** | | | |
| Numerical stability term | $\epsilon$ | Denominator stabilizer for relative change | $1 \times 10^{-8}$ |
| Temperature Scaling | $T$ | Temperature for gradient scoring | 0.07 |

### A.3.5 ABLATION EXPERIMENTS AND THEORETICAL ANALYSIS

To clarify the contribution of each component and provide a deeper theoretical analysis of our framework, we conducted a comprehensive ablation study. We present the detailed data from our ablation studies in Table 4. These figures correspond to the results shown in Figure 6.

Table 4: **Ablation Experiments.** Performance comparison of NeuroAlign (Full) and its variants on THINGS-EEG2 and THINGS-MEG datasets. The values in parentheses indicate the performance drop compared to the full model.

| Dataset | Method | Intra-Subject | | Inter-Subject | |
|---|---|---|---|---|---|
| | | Top-1 | Top-5 | Top-1 | Top-5 |
| THINGS-EEG2 | **NeuroAlign (Full)** | **48.1** | **78.1** | **14.5** | **36.4** |
| | w/o VSE | 42.3 ($\downarrow$5.8) | 74.8 ($\downarrow$3.3) | 12.3 ($\downarrow$2.2) | 33.5 ($\downarrow$2.9) |
| | w/o SGA | 39.5 ($\downarrow$8.6) | 73.0 ($\downarrow$5.1) | 13.6 ($\downarrow$0.9) | 35.8 ($\downarrow$0.6) |
| | w/o DLA | 24.7 ($\downarrow$23.4) | 58.4 ($\downarrow$19.7) | 10.2 ($\downarrow$4.3) | 27.5 ($\downarrow$8.9) |
| THINGS-MEG | **NeuroAlign (Full)** | **26.5** | **53.3** | **3.9** | **13.1** |
| | w/o VSE | 25.0 ($\downarrow$1.5) | 51.7 ($\downarrow$1.6) | 3.4 ($\downarrow$0.5) | 11.4 ($\downarrow$1.7) |
| | w/o SGA | 21.9 ($\downarrow$4.6) | 51.5 ($\downarrow$1.8) | 2.6 ($\downarrow$1.3) | 10.5 ($\downarrow$2.6) |
| | w/o DLA | 16.8 ($\downarrow$9.7) | 44.0 ($\downarrow$9.3) | 1.8 ($\downarrow$2.1) | 9.0 ($\downarrow$4.1) |

Our study validates that NeuroAlign's success is rooted in simulating the brain's dual-stream processing. This is achieved not by simply combining components, but through a dynamic integration. The DLA orchestrates a phased learning curriculum between our two pathways: the bottom-up VSE module and the top-down SGA module. It first prioritizes the VSE to establish a stable perceptual grounding by aligning EEG with salient visual features. Once this base is established, it progressively shifts focus to the SGA, which uses semantic priors to guide these representations toward a high-level semantice space. This strategy resolves the inherent instability of directly aligning noisy brain signals with abstract concepts, creating a more effective and stable learning process.

The ablation results provide strong empirical evidence for this theory in Table 4. While removing the VSE or SGA modules leads to respective Top-1 accuracy drops of 5.8% and 8.6%, ablating the DLA module triggers a 23.4% performance degradation. This large decline confirms that the model's success stems not from a simple summation of its parts, but from their dynamic synergy.

### A.3.6 LARGE LANGUAGE MODEL ANALYSIS

To investigate how the quality of text descriptions impacts cross-modal alignment, we evaluated descriptions generated by three different LLMs (BLIP-2, LLAVA-1.5(Liu et al., 2024b), and QWEN-2(Team et al., 2024)), each using three distinct prompts. The results are shown in Figure 9 and Table 5.

Table 5: **Performance Comparison with Different Text Priors.** The baseline methods (No Text, Label Only) do not utilize LLMs for prompt generation, thus a single reference score is shown. For LLM-based priors, we report accuracy across three different source models.

| Setting | Prompt Description | Accuracy (Mean $\pm$ Std) | | |
|---|---|---|---|---|
| ***Baselines (No LLM generation)*** | | | | |
| w/o Text Prior | No text | 40.1 $\pm$ 0.9 | | |
| Label Only | Class labels (e.g., "eagle") | 45.6 $\pm$ 1.3 | | |
| ***With LLM Prior*** | | BLIP2 | LLAVA1.5 | QWEN2 |
| Prompt A | Concise: "This is a photo of..." | **48.1 $\pm$ 0.8** | **48.5 $\pm$ 0.9** | **47.4 $\pm$ 1.3** |
| Prompt B | Object-focused: "Describe the main object in the picture" | 47.5 $\pm$ 1.8 | 47.6 $\pm$ 1.2 | 47.2 $\pm$ 2.4 |
| Prompt C | Detailed: "Briefly describe the 'label' in the images." | 46.6 $\pm$ 2.3 | 44.8 $\pm$ 2.1 | 45.9 $\pm$ 2.6 |

**Text priors are crucial and superior to simpler labels.** Compared with the results using text priors which Top-1 accuracy is 48.1%, without text priors the Top-1 accuracy is 40.1% (-8%). Even when only label information is used, the accuracy is 5.5% higher than the baseline without text. So we can conclude that, LLMs provide more constrained and detailed descriptions of the core object. This more effectively activates and guides the image features to match the brain's high-level semantic priors, significantly improving retrieval accuracy.

**Performance is robust across different LLMs with effective prompts.** For straightforward prompts like A and B, all three LLMs perform similarly and show comparable results. However, when the prompt required the LLM to generate more complex and detailed descriptions (as with Prompt C), the differences in how each LLM handled these instructions became more apparent. This indicates that a simple, direct prompt is more consistent performance regardless of the specific language model used.

**Concise prompts outperform detailed ones.** Our analysis clearly shows that the prompting strategy is a critical factor. Using LLAVA-1.5 as an example, the concise Prompt A achieved the highest accuracy (48.5%), a significant 3.7% improvement over the detailed Prompt C (44.8%). The success of Prompt A lies in its ability to generate direct descriptions focused on the core object. Conversely, detailed prompts like C tend to introduce excessive or irrelevant information (e.g., "a large bird of prey," "keen eyesight"), which acts as semantic noise. This noise interferes with the alignment process, weakening the model's ability to focus on core visual information.

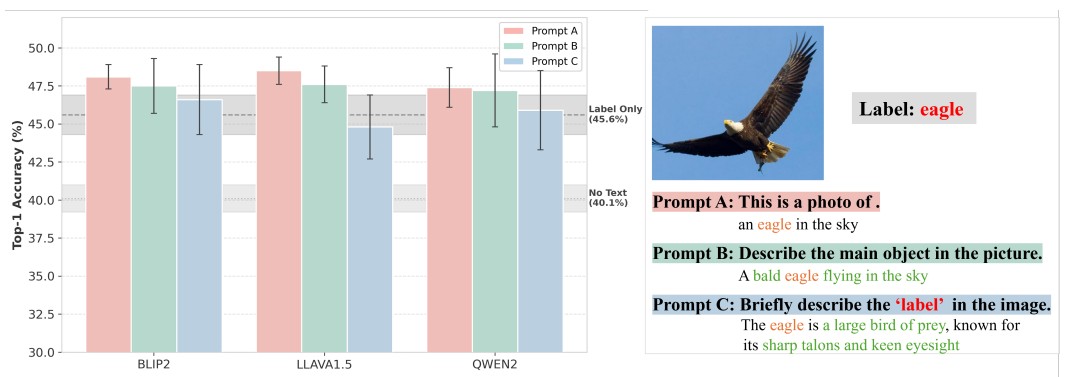

Figure 9: **The performance of different LLMs: BLIP-2, LLAVA-1.5, QWEN-2 and their three prompts of Top-1 accuracy on the THINGS-EEG2. Left figure**: The baseline performance without text priors and with only label are compared. The error bars indicate the sensitivity of the model to different LLMs and prompts. **Right figure**: Prompts A, B, C, where red text represents the image category, orange text represents the main object, and green text represents detailed descriptive information.

### A.3.7 SPATIAL CONTRIBUTION ANALYSIS OF EEG CHANNELS

To further demonstrate the biological plausibility of the learned EEG representations, we analyzed the spatial dynamics of feature contributions by quantifying the impact of specific EEG channels on the model's inference process (Shrikumar et al., 2017). Figure 10 visualizes the topographic distribution of these contributions, derived by averaging all test trials across the 10 subjects.

Across all participants, a distinct and dominant response was observed in the posterior brain regions. This activity was specifically concentrated in the occipital and posterior-parietal areas. The intensity of this activation varied among subjects. For example, Subject 1 showed stronger activation than Subject 4, which can be attributed to individual differences in Signal-to-Noise Ratio (SNR) or skull conductivity. Despite these variations, the posterior-dominant activation pattern remained remarkably consistent across all participants.

This spatial distribution is highly consistent with the functional anatomy of the human visual system, indicating our model successfully learned to prioritize signals from the primary visual cortex (the

region responsible for processing visual input). The model also effectively suppressed irrelevant noise from frontal or temporal regions. This result confirms that the model's high classification performance is driven by biologically valid visual processing signals.

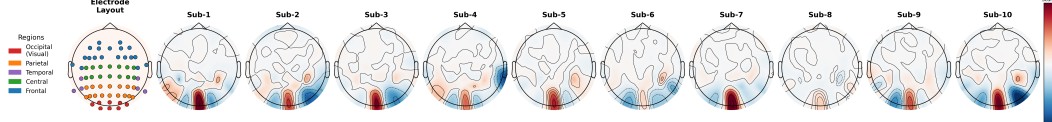

Figure 10: **Topographic visualization of EEG feature contributions.** The leftmost panel is standard EEG electrode layout. Feature saliency topographic maps for 10 individual subjects. Color saturation is proportional to feature importance. The maps consistently show that the most contributory features are concentrated in posterior brain regions, primarily the occipital and parietal lobes, aligning with the primary functional areas for visual processing.

### A.3.8 RETRIEVAL CASE ANALYSIS

This section presents the Top5 retrieval performance of our model on multicategory objects from the THINGSEEG2 dataset, including both successful and unsuccessful cases as shown in Figure 11 and Figure 12, respectively.

As observed in Figure 11, the model accurately identifies core object features and retrieves highly relevant results. Notably, it not only recognizes categorical attributes but also captures finegrained visual characteristics such as color, texture, and shape. These results suggest that the model develops a meaningful semantic understanding and establish reasonable associations between related objects.

However, the failure cases in Figure 12 indicate that the model tends to confuse objects with similar shapes or colors. For example, associating coffee cups with other circular objects or incorrectly recognizing animal species. This behavior reflects a tendency to rely on superficial visual similarities rather than highlevel semantics. The findings imply that while EEG signals convey visualperceptual information, the encoded semantics remain relatively shallow, primarily anchored at the level of basic visual features.

These failure cases reveal a fundamental challenge in neural signal decoding: **how to extract sufficiently rich and accurate high-level semantic information from brain activity that is inherently noisy and information-sparse**? This insight directs future work toward enhancing the semantic understanding capabilities of EEG-based decoding algorithms, with a focus on strengthening semantic representation learning.

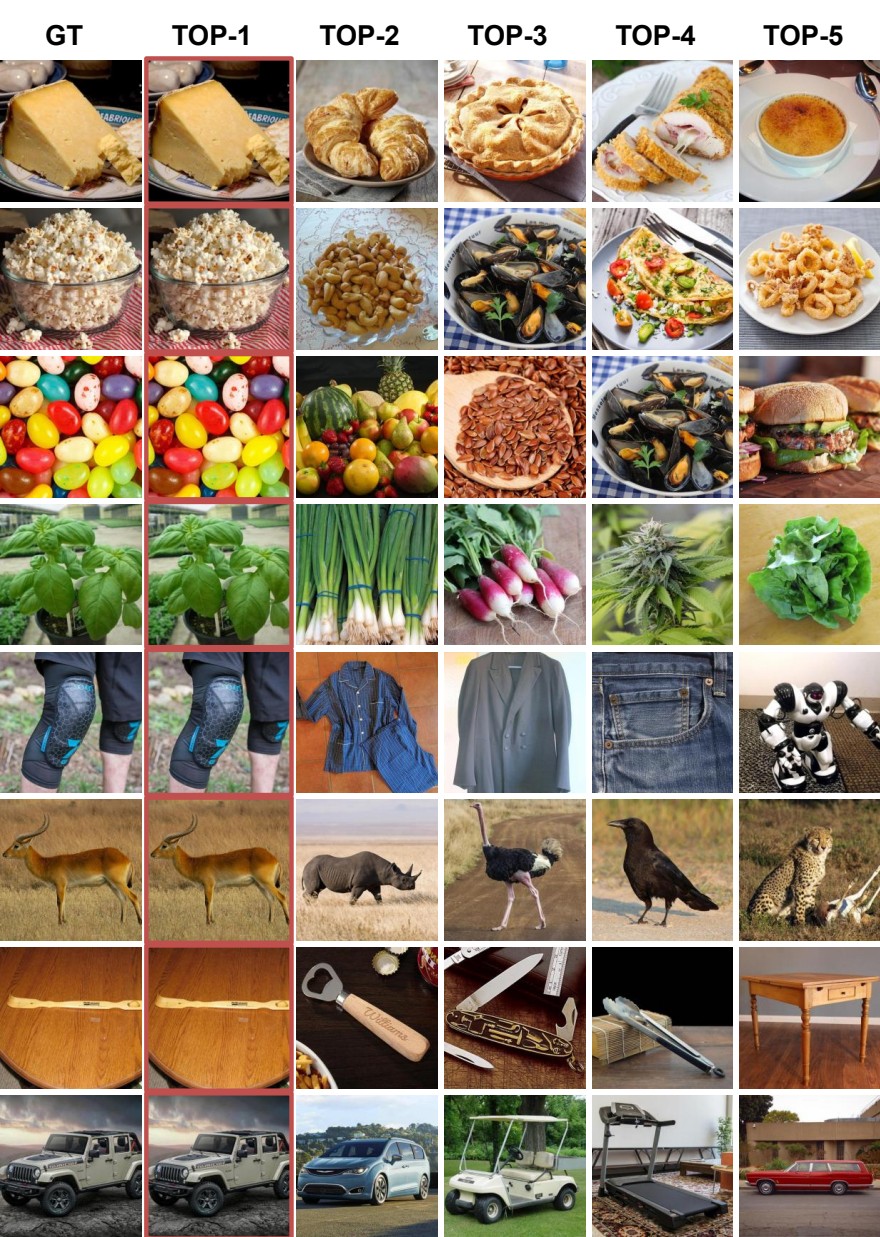

Figure 11: Successful cross-modal retrieval examples on the THINGS-EEG2 dataset. The results demonstrate accurate EEG-image alignment and the model's ability to capture high-level semantic concepts alongside fine-grained visual attributes.

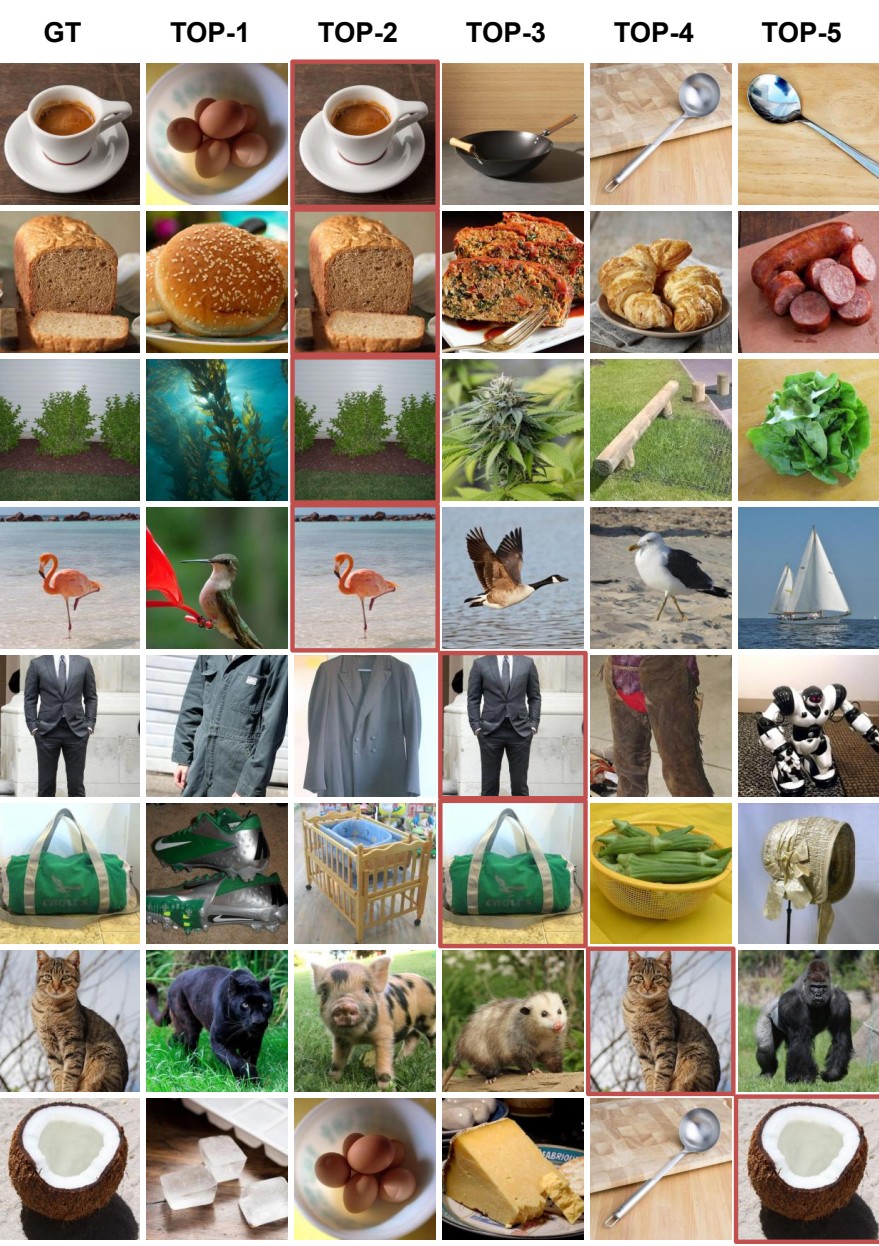

Figure 12: Examples of retrieval failures on THINGS-EEG2, illustrating the model's preference for low-level visual features (shape, color) rather than semantic content. This confusion indicates the current approach's limitation in decoding high-level semantics from EEG.

