# OpenReview forum: "A Neuroscience-inspired Framework for Tri-modality Alignment of Brain Signals, Vision, and Language"
_ICLR.cc/2026/Conference — Submitted to ICLR 2026_

### Official Review · Reviewer_LVmN · 2025-10-24

**Soundness:** 4
**Presentation:** 3
**Contribution:** 3
**Rating:** 6
**Confidence:** 3

**Summary:**

This paper introduces NeuroAlign, a neuroscience-inspired framework for aligning brain (EEG), vision, and language modalities to improve visual retrieval from brain signals. It mimics the brain’s bottom-up and top-down visual processing through a Visual Saliency Extraction module for physiologically consistent image features, a Semantic Guidance Alignment module using LLM-generated text priors, and a Dynamic Loss Adjustment mechanism for adaptive multi-objective learning. Experiments on THINGS-EEG2 and THINGS-MEG datasets show that NeuroAlign achieves state-of-the-art zero-shot retrieval performance, significantly outperforming previous brain-vision alignment models. Overall, the framework bridges neuroscience and multimodal learning, offering both stronger decoding accuracy and greater interpretability of brain–vision–language correspondence.

**Strengths:**

1. The tri-modality setup (EEG–vision–language) with KL-based semantic guidance is methodologically clear and addresses specific bottlenecks in prior EEG decoding approaches (e.g., NICE, BraVL, UBP).

2. NeuroAlign consistently surpasses recent baselines on both intra- and inter-subject retrieval across two datasets. The inter-subject improvement, though modest, is statistically meaningful given the difficulty of EEG generalization.

3. The analysis (temporal saliency, t-SNE category clustering) convincingly demonstrates alignment between neural and visual-semantic representations. The results support the claim that NeuroAlign better mirrors real cortical processing dynamics.

**Weaknesses:**

1. The method’s dependence on LLM-generated captions may raise reproducibility and stability issues. It is unclear how much the results rely on specific text priors versus inherent EEG-vision correlations.

2. The work focuses heavily on visual saliency and semantic analysis but provides little insight into which EEG features drive cross-modal alignment. Some interpretability analysis (e.g., channel/time contribution) would improve clarity.

3. Regarding the dynamic loss adjustment component, the authors are encouraged to provide the curve of the adjustable hyperparameter during the training to see how it learns during the optimization.

4. The analysis section lacks of in-depth analysis towards the theoretical contribution of this work. The authors are suggested to enrich the description in the analysis section and provide more information regarding why their proposed method works better.

5. The proposed method consists of three major components, i.e., VSE, SGA, and DLA. What are the relationship among these components? Can these components help each other?

6. In the experiment section, the authors are encouraged to add more ablation experiments of the proposed method. VSE, SGA, and DLA should be removed one-by-one to validate their individual performance contributed to the NeuroAlign model.

**Questions:**

1. How sensitive is NeuroAlign’s performance to the specific choice of LLM-generated captions?

2. Have the authors evaluated whether different captioning models or prompting strategies affect the alignment results and reproducibility?

3. Can the model still perform well when no textual priors or simpler labels are used?

4. Which EEG channels or temporal segments contribute most to the cross-modal alignment?

5. Could the authors provide visualizations or attention maps showing how EEG features correspond to visual or semantic elements?

6. Could the authors provide the evolution curves of the adaptive weights (w1, w2, w3) during training to illustrate how the model balances objectives over time?

7. How stable is the DLA mechanism across different random seeds or subjects? Does DLA always converge to similar weighting behavior across runs?

8. What theoretical insights explain why integrating VSE, SGA, and DLA leads to better cross-modal alignment? Have the authors conducted ablation studies by removing VSE, SGA, or DLA individually to quantify their individual contributions?

---

> ### Author Response · Authors · 2025-11-27
> **Response to Reviewer LVmN Part (1/3)**
>
> We would like to thank the reviewer’s for your time and valuable comments. We have carefully considered all the suggestions and revised the paper accordingly. The point-by-point responses are listed below.
>
> ---
>
> >*W1: The method’s dependence on LLM-generated captions may raise reproducibility and stability issues. It is unclear how much the results rely on specific text priors versus inherent EEG-vision correlations.*
>
> >*Q1: How sensitive is NeuroAlign’s performance to the specific choice of LLM-generated captions?*
>
> >*Q2: Have the authors evaluated whether different captioning models or prompting strategies affect the alignment results and reproducibility?*
>
> >*Q3: Can the model still perform well when no textual priors or simpler labels are used?*
>
> *Response to W1, Q1, Q2 and Q3*: Thanks for your questions.
>
> We conduct a **comprehensive analysis** evaluating **different LLMs**, **prompting strategies**, and the necessity of text priors, details are presented in Appendix A.3.6 (line 1062). Here we summarized the results in the table below.
>
> | Method / Setting | Prompt Strategy || Top-1 Accuracy||
> | :--- | :--- | :---: | :---: | :---: |
> | **w/o Text Prior** | No text || 40.1 ± 0.9 |  |
> | **Label Only** | Using fixed class labels (e.g., "eagle") | |45.6 ± 1.3 |  |
> | **With LLM Prior** | | **BLIP2** | **LLAVA1.5** | **QWEN2** |
> | &emsp;Prompt A | Concise: "This is a photo of." | 48.1 ± 0.8 | 48.5 ± 0.9 | 47.4 ± 1.3 |
> | &emsp;Prompt B | Object-focused: "Describe the main object..." | 47.5 ± 1.8 | 47.6 ± 1.2 | 47.2 ± 2.4 |
> | &emsp;Prompt C | Detailed: "Briefly describe the 'label'..." | 46.6 ± 2.3 | 44.8 ± 2.1 | 45.9 ± 2.6 |
>
> (1) *(Addresses W1 & Q3)* **Text Priors are Crucial and Superior to Simpler Labels.** Using LLM-generated priors (e.g., 48.1% Top-1) significantly outperforms baselines using no text prior (40.1%, - 8.0%) and simpler class labels (45.6%, a drop of 2.5%).
>
> (2) *(Addresses W1, Q1 & Q2)* **Performance is Robust Across Different LLMs with Effective Prompting.** Regarding the choice of LLM, our results show high stability and comparable performance across different models (BLIP-2, LLAVA-1.5 [1], QWEN-2 [2]) when using effective, concise prompts. With Prompt A, the Top-1 accuracies are **48.1%**, **48.5%**, and **47.4%** respectively. This directly addresses the reproducibility concern, indicating that the framework's success is not dependent on a specific LLM.
>
> (3) *(Addresses Q2)* **Concise Prompts Outperform Detailed Ones:** Our analysis reveals a clear preference for concise, object-focused descriptions. For instance, with LLAVA-1.5, the concise Prompt A (**48.5%**) outperforms the object-focused Prompt B (**47.6%**) and significantly surpasses the detailed Prompt C (**44.8%**). This suggests that overly detailed or redundant text can introduce semantic noise, which undermines the guidance mechanism.
>
> ---
>
> >*W2: The work focuses heavily on visual saliency and semantic analysis but provides little insight into which EEG features drive cross-modal alignment. Some interpretability analysis (e.g., channel/time contribution) would improve clarity.*
>
> >*Q4: Which EEG channels or temporal segments contribute most to the cross-modal alignment?*
>
> >*Q5: Could the authors provide visualizations or attention maps showing how EEG features correspond to visual or semantic elements?*
>
> *Response to W2, Q4, and Q5*: Thanks for your questions. We conduct a detailed temporal and spatial contribution analysis in Section 4.2.1 (line 378) and Appendix A.3.7 (line 1121), respectively. The key findings presented below.
>
> (1) *(Addresses W2 & Q4)* **Temporal Contribution Analysis:** We evaluate two cumulative window types: a forward-increasing window [0, t] and a backward-decreasing window [t, 1000] (shown in Figure 3(a) and (b)). Our results consistently reveal that the early visual processing stage (approximately **100-500ms** post-stimulus) contains the most critical neural information for cross-modal retrieval, yielding the highest accuracy.
>
> (2) *(Addresses W2, Q4 & Q5)* **Spatial Contribution and Visualization:** We generate topographic saliency maps by averaging all test trials across the 10 subjects (shown in Figure 10) [3]. The most influential features consistently originate from the occipital and posterior-parietal lobes, aligning perfectly with the functional anatomy of the primary and secondary visual cortices. This activation pattern is remarkably robust across all subjects. This demonstrates that our model has learned a generalizable and biologically valid feature representation.

---

> ### Author Response · Authors · 2025-11-27
> **Response to Reviewer LVmN Part (2/3)**
>
> >*W3: Regarding the dynamic loss adjustment component, the authors are encouraged to provide the curve of the adjustable hyperparameter during the training to see how it learns during the optimization.*
>
> >*Q6: Could the authors provide the evolution curves of the adaptive weights (w1, w2, w3) during training to illustrate how the model balances objectives over time?*
>
> >*Q7: How stable is the DLA mechanism across different random seeds or subjects? Does DLA always converge to similar weighting behavior across runs?*
>
> *Response to W3, Q6, and Q7*: Thanks for your questions.
>
> We provide the evolution curves of the adaptive weights ($w_1, w_2, w_3$) in Section 4.2.3 (line 450) and Appendix A.2.3 (line 882). Our analysis confirms that the DLA mechanism exhibits **highly stable and consistent convergence** behavior, and we verified through two targeted experiments:
>
> (1) *(Addresses W3 & Q7)* **Stability Across Random Seeds:** To assess reproducibility, we train our model on a single subject's data (Subject 8) using five different random seeds. As shown in the main paper (Figure 5(b), line 469), the resulting weight evolution curves are nearly identical, with a minimal standard deviation (**< 0.1**). This demonstrates that the convergence of DLA is highly deterministic and robust to random initialization, directly addressing the stability concern.
>
> (2) *(Addresses W3 & Q7)* **Consistency Across Subjects:** To evaluate generalizability, we analyze the weight trajectories for all 10 subjects. As detailed in Appendix A.2.3 (line 882), despite natural variations in individual data, the curves exhibit a remarkably consistent pattern across all subjects. This consistency proves that DLA captures a generalizable and cognitively-plausible learning progression, rather than overfitting to subject-specific artifacts.
>
> (3) *(Addresses W3 & Q6)* **Cognitively-Plausible Learning Progression:** In all experiments, the DLA weights consistently follow a two-stage, cognitively-plausible trajectory:
>
> * **Early Stage:** The model prioritizes low-level Image-EEG alignment (higher $w_1$), simulating a stimulus-driven process.
>
> * **Later Stage:** As training progresses, the weight for semantic guidance (higher $w_3$) steadily increases, reflecting a shift towards a semantically-guided process, before reaching a stable state.

---

> ### Author Response · Authors · 2025-11-27
> **Response to Reviewer LVmN Part (3/3)**
>
> >*W4: The analysis section lacks of in-depth analysis towards the theoretical contribution of this work. The authors are suggested to enrich the description in the analysis section and provide more information regarding why their proposed method works better.*
>
> >*W5:The proposed method consists of three major components, i.e., VSE, SGA, and DLA. What are the relationship among these components? Can these components help each other?*
>
> >*W6: In the experiment section, the authors are encouraged to add more ablation experiments of the proposed method. VSE, SGA, and DLA should be removed one-by-one to validate their individual performance contributed to the NeuroAlign model.*
>
> >*Q8: What theoretical insights explain why integrating VSE, SGA, and DLA leads to better cross-modal alignment? Have the authors conducted ablation studies by removing VSE, SGA, or DLA individually to quantify their individual contributions?*
>
> *Response to W4, W5, W6 and Q8:* Thank you for your question! We include the requested ablation studies in Section 4.2.4 (line 486) and Appendix A.3.5 (line 1026), and we provide a unified theoretical and experimental analysis below.
>
> (1) *(Addresses W4 & Q8)* The core theoretical insight of NeuroAlign is that effective Brain-Image retrieval under the RSVP paradigm requires **a model that simulates the brain's dual-stream visual processing mechanism.** Our framework addresses this by explicitly modeling two synergistic pathways and their dynamic interplay:
>
> *  **Bottom-Up Stimulus-Driven Pathway (VSE):** Under RSVP, early EEG signals are dominated by low-level visual saliency (e.g., luminance, edges) [4]. Our VSE module simulates this pathway by forcing the model to first ground its representations in salient visual features that are physiologically consistent with the initial neural response.
>
> *  **Top-Down Task-Driven Pathway (SGA):** To form high-level semantics, the brain leverages prior knowledge to interpret perceptual signals [5]. Our SGA module simulates this by using an LLM as an external stable semantic priors, guiding the noisy EEG features towards a consistent semantic space.
>
> (3) *(Addresses W5)* The components are not independent but form a synergistic system, with the DLA module coordinating their interactions. Their relationship is as follows:
>
> *  **VSE provides a grounded foundation:** It first aligns EEG with low-level, salient image features.
>
> *  **SGA builds upon this foundation:**  It then uses this initial alignment to guide the features towards a higher-level semantic meaning.
>
> *  **DLA dynamically mediates this process:**  In the early training stages, DLA prioritizes the VSE-driven alignment to establish a stable perceptual base. As this alignment matures, DLA progressively increases the influence of SGA, shifting the model's focus from "what it sees" to "what it means." This dynamic mediation ensures that each pathway contributes most effectively at the appropriate stage of learning.
>
> (4) *(Addresses W6 & Q8)* We present ablation studies in Appendix A.3.5 (line 1014), and summarized in the table:
>
> | Dataset | Method | Intra-Subject(Top-1) | Intra-Subject(Top-5) | Inter-Subject(Top-1) | Inter-Subject(Top-5) |
> | :--- | :--- | :---: | :---: | :---: | :---: |
> | **THINGS-EEG2** | **NeuroAlign (Full)** | **48.1** | **78.1** | **14.5** | **36.4** |
> | | w/o VSE | 42.3 (↓5.8) | 74.8 (↓3.3) | 12.3 (↓2.2) | 33.5 (↓2.9) |
> | | w/o SGA | 39.5 (↓8.6) | 73.0 (↓5.1) | 13.6 (↓0.9) | 35.8 (↓0.6) |
> | | w/o DLA | 24.7 (↓23.4) | 58.4 (↓19.7) | 10.2 (↓4.3) | 27.5 (↓8.9) |
> | **THINGS-MEG** | **NeuroAlign (Full)** | **26.5** | **53.3** | **3.9** | **13.1** |
> | | w/o VSE | 25.0 (↓1.5) | 51.7 (↓1.6) | 3.4 (↓0.5) | 11.4 (↓1.7) |
> | | w/o SGA | 21.9 (↓4.6) | 51.5 (↓1.8) | 2.6 (↓1.3)
>
> Removing any single component degrades performance. The **largest performance drop occurs when DLA is ablated**. This confirms that **the dynamic integration is the most critical element**, as it maximizes the synergy between the VSE and SGA pathways. This provides strong evidence that our model's success stems from the neuroscience-inspired design.
>
> We believe that our responses and revisions have addressed all the concerns raised. Thank you again for your consideration.
>
> ---
>
> [1] Liu H, et al. Improved baselines with visual instruction tuning. CVPR 2024.
>
> [2] Team Q. Qwen2 technical report. arXiv preprint 2024.
>
> [3] Shrikumar A, et al. Learning important features through propagating activation differences. ICML 2017.
>
> [4] Cichy R M, et al. Resolving human object recognition in space and time. Nature neuroscience 2014.
>
> [5] VanRullen R. The power of the feed-forward sweep. Advances in Cognitive Psychology 2008.

---

> > ### Comment · Reviewer_LVmN · 2025-11-27
> > **Response to the authors**
> >
> > Thank you very much for your rebuttal, which has addressed most of my concerns. I appreciate the effort you put into clarifying the points I raised.
> >
> > I have one remaining question regarding Fig. 5(a): could you clarify what the small circles represent? At the moment, the number of circles does not seem to match the number of runs you referred to in the rebuttal.
> >
> > It would be helpful if you could explain more clearly what these circles denote and add the performance values of the individual runs directly to the figure. I believe this would make it easier for readers to understand the upper-bound performance your method can achieve.
> >
> > If this issue is addressed, I will be happy to raise my score to 8.

---

> > > ### Author Response · Authors · 2025-11-28
> > > **Response to Reviewer LVmN**
> > >
> > > >*Q9: could you clarify what the small circles represent? At the moment, the number of circles does not seem to match the number of runs you referred to in the rebuttal.*
> > >
> > > *Response to Q9:* Thank you for your positive feedback and this excellent follow-up question.
> > >
> > > (1) To clarify, the circles in the original figure were **statistical outliers**, which is why their number did not match the total number of subjects. In our experiment, these outliers represent subjects with exceptionally high or low accuracy compared to the rest of the group.
> > >
> > > (2) We fully agree with your suggestion, and **revise Figure 5(a) in the revised paper (line 468)**. The updated figure now overlays all 10 individual subject data points, including the performance range and upper bounds for each method.
> > >
> > > We hope this revision fully addresses your concern and appreciate your valuable feedback.

---

> > > > ### Comment · Reviewer_LVmN · 2025-11-28
> > > > **Response**
> > > >
> > > > Dear Authors; Dear AC
> > > >
> > > > @ Authors, thank you for your comment. I have no other concern. However it seems that the final score editting is blocked.
> > > >
> > > >  @AC I want to raise my rating to 8 since the authors have addressed most of my concerns and provided really good rebuttal. Could you take 8 as my final score since the system can not edit the final rating? Thank you.

---

> > > > > ### Author Response · Authors · 2025-11-28
> > > > > **Replying to Official comment by Reviewer LVmN**
> > > > >
> > > > > Thanks very much for your reply and recognition. We are happy to see that your concerns have been addressed.

---

### Official Review · Reviewer_A8NT · 2025-10-31

**Soundness:** 2
**Presentation:** 3
**Contribution:** 2
**Rating:** 4
**Confidence:** 4

**Summary:**

This paper introduces NeuroAlign, a neuroscience-inspired framework for visual retrieval from brain signals. It argues that prior approaches suffer from three issues:  feature–physiology mismatch, weak intra-modality semantic consistency and reliance on static image–text spaces that lack dynamic semantic priors. NeuroAlign addresses these by first extract salient object visual features and align them with EEG  embedding. It then conducts semantic guidance alignment and dynamic loss adjustment to achieve dynamic brain-to-vision alignment.

**Strengths:**

- This paper is well-written, with three motivations clearly illustrated.
- The experiments demonstrate the effectiveness over the baseline model.

**Weaknesses:**

- From the introduction, the authors aim to align EEG embeddings with low-level features. In lines 205–206, they mention that “directly aligning EEG signals with raw RGB images often leads to overfitting on high-frequency details,” citing UBP. However, high-frequency details actually correspond to low-level features, and UBP is designed to remove these components, which appears contradictory to the approach taken in this paper.

- The methodology and motivation of Center-surround Antagonism part is not well-explained.

- Some key ablation studies are missing. What’s the decoding accuracy when ablating the semantic guidance alignment? From Figure 6, the VSE module visualization does not clearly demonstrate enhanced attention to specific image regions. How does it compare with UBP, which applies the Fovea Blur mechanism? Additional experimental analysis is recommended.
- In Figure 1, several anatomical terms such as PFC, LGN, and PPC are presented. What is their specific contribution to explaining the core principle of the proposed method? Please clarify their relevance to the framework. In addition, the design appears quite similar to [1]; this work should be properly cited.

[1] Choi M, Han K, Wang X, et al. A dual-stream neural network explains the functional segregation of dorsal and ventral visual pathways in human brains[J]. Advances in Neural Information Processing Systems, 2023, 36: 50408-50428.

**Questions:**

- In the main experiments on the EEG dataset, how was the intra-subject result of UBP obtained? The reported Avg acc do not match the official benchmark (Top-1 = 50.9, Top-5 = 79.7), which are notably higher.
- What does “low-level neural responses” in introduction section refer to? Does it mean the low-level components within the neural signals themselves, or the neural responses elicited by low-level visual stimuli?
- What’s the meaning of E_d in Equation 1.  Besides, the parentheses in Equation 1 are mismatched.
- Equation (5) seems to contain a possible typo: the image representation should be denoted as v rather than t.

Please refer to weaknesses for other questions.

---

> ### Author Response · Authors · 2025-11-27
> **Response to Reviewer A8NT Part (1/2)**
>
> We appreciate the reviewer’s insightful feedback and carefully revise the paper according to your suggestions. All revisions are highlighted in blue. Our point-by-point responses are provided below.
>
> ---
>
> >*W1: …However, high-frequency details actually correspond to low-level features, and UBP is designed to remove these components, which appears contradictory to the approach taken in this paper.*
>
> *Response to W1*: Thanks for your question. High-frequency details do indeed correspond to low-level features,.
> *  In our paper, low-level features refer to **salient features** (e.g., edges, colors) that trigger neural responses in RSVP tasks. We achieve a guided alignment by extracting salient features.
> *  In UBP, low-level features refer to **irrelevant high-frequency noise.**
>
> ---
>
> >*W2: The methodology and motivation of Center-surround Antagonism part is not well-explained.*
>
> *Response to W2*: Thanks for your question, and we apologize for any confusion resulting from our original description. We refine the motivation and method descriptions in the revised paper (line 228), and provide a detailed implementation in Appendix A.2.1 (line 803).
>
> (1) **Motivation:** The center-surround antagonism mechanism is directly inspired by its function in the receptive fields of neurons in the retina and visual cortex, implemented via a multi-scale Difference of Gaussians (DoG) filter as modeled by [1]. **We simulate how the visual system enhances salient features like edges while suppressing irrelevant backgrounds.** This process obtains a simplified, saliency-focused image representation that better reflects the brain's initial stimulus-driven neural response.
>
> (2) **Mathematical Modeling using Gaussian Filters:** For each pixel, we compute the difference between the responses of two Gaussian filters: a narrow 'center' filter $\mathcal{G}_c$ and a wider 'surround' filter $\mathcal{G}_s$.
>
> $
>     \mathcal{G}_c(x, y, s) = \frac{1}{2\pi\sigma_c^2} \exp \left( -\frac{x^2 + y^2}{2\sigma_c^2} \right)
> $
>
> $
>     \mathcal{G}_s(x, y, s) = \frac{1}{2\pi\sigma_s^2} \exp \left( -\frac{x^2 + y^2}{2\sigma_s^2} \right)
> $
>
> To ensure robustness to variations in object size and distance, this DoG operation is performed across multiple scales $N_s$, and the results are aggregated to produce the final antagonistic response map $R_a$.
>     $$
>     R_a = \mathcal{N} \left( \left| \frac{1}{N_s} \sum_{i=1}^{N_s} \left[ \mathcal{G}_c(S_a) - \lambda \mathcal{G}_s(S_a) \right]_i \right|^\gamma \right)
>     $$
>
> ---
>
> >*W3: Some key ablation studies are missing…Additional experimental analysis is recommended.*
>
> *Response to W3*: Thanks for your question.
>
> (a) The relevant **ablation experiments** are presented in **Section 4.2.4 (line 486)** and **Appendix A.3.5 (line 1026)** of the paper. Here we present the specific results with and without SGA in the table below.
>
> | Dataset         | Intra-subject (Top-1)     | Intra-subject (Top-5)     | Inter-subject (Top-1)     | Inter-subject (Top-5)     |
> | :-------------- | :-----------------------: | :-----------------------: | :-----------------------: | :-----------------------: |
> | **THINGS-EEG2** | 39.5 / **48.1** (↑8.6)  | 73.0 / **78.1** (↑5.1)  | 21.9 / **26.2** (↑4.3)  | 51.5 / **53.3** (↑1.8)  |
> | **THINGS-MEG**  | 13.6 / **14.5** (↑0.9)  | 35.8 / **36.4** (↑0.6)  | 2.6 / **3.9** (↑1.3)    | 10.5 / **13.1** (↑2.6)  |
>
> Across all datasets and settings, the SGA module consistently improved retrieval accuracy. Notably, in the THINGS-EEG2 intra-subject task, SGA boosted the Top-1 accuracy from **39.5%** to **48.1%**.
>
> (b) We quantitatively **compare the performance of the VSE module and the Fovea Blur mechanism**. The results are shown below:
>
> | Model                     | Intra-subject (Top-1) | Intra-subject (Top-5) | Inter-subject (Top-1) | Inter-subject (Top-5) |
> | :------------------------ | :-------------------: | :-------------------: | :-------------------: | :-------------------: |
> | UBP                     | 45.7                  | 76.6                  | 11.9                  | 32.1                  |
> | Ours(w/o VSE)+FoveaBlur | 45.2                  | 77.2                  | 11.8                  | 33.8                  |
> | **Ours**                | **48.1**              | **78.1**              | **14.5**              | **36.4**              |
>
> Our full model with the VSE module outperforms both the UBP baseline and our model variant using Fovea Blur, particularly in the more challenging inter-subject setting where it achieves a Top-1 accuracy of 14.5%, compared to 11.9% (UBP) and 11.8% (Fovea Blur).

---

> ### Author Response · Authors · 2025-11-27
> **Response to Reviewer A8NT Part (2/2)**
>
> >*W4: In Figure 1, several anatomical terms such as PFC, LGN, and PPC are presented. … Please clarify their relevance to the framework. In addition, the design appears quite similar to [1]; this work should be properly cited.*
>
> *Response to W4*: Thanks for your questions. We add extra details for Figure 1 (line 73) and citation to Choi [2] (line 53) in the revised paper.
>
> (a) The three anatomical terms that appear in Figure 1 (line 54) : PPC, PFC, and LGN are included to **more comprehensively present the visual dual-stream architecture that our framework is based on.**
>
> *  The Visual Saliency Extraction (VSE) module to simulate the bottom-up processing from "Retina->LGN->V1," extracting features of salient visual areas in brain signals under the RSVP paradigm.
>
> *  The Semantic Guided Alignment (SGA) module and the Dynamic Loss Adjustment (DLA) mechanism to simulate the top-down attention modulation process driven by "PFC->PPC." This allows us to dynamically adjust the alignment between brain signals and images using prior knowledge.
>
> (b) The content cited in th paper is:“Some works like (Choi et al., 2023) are also inspired by the dual-stream model, but they focus on encoding fMRI signals to model visual behavior, while ours is on decoding EEG/MEG signals for image retrieval, which presents unique challenges due to the lower signal-to-noise ratio and high temporal dynamics of EEG signals.”
>
> ---
>
> >*Q1: In the main experiments on the EEG dataset, how was the intra-subject result of UBP obtained? …*
>
> *Response to Q1*: Thanks for your question. It allows us to our evaluation protocol and correct an oversight. We update Table 1 (line 325) in the revised paper.
>
> To ensure a fair comparison, all baselines are **using same 63 EEG channels** and data-split settings. The data preprocessing and splitting methods are detailed in Appendix A.3.2 and A.3.3. UBP uses 17 vision-relevant channels, resulting different datas in our paper.
>
> In the inter-subject results, we mistakenly reported the official UBP values instead of the ones we got using 63 channels. The updated UBP’s average performance in Table 1 is now **11.9%** (Top-1) and **32.1%** (Top-5).
>
> We apologize for the oversight, it was never our intention to selectively report results to appear better than others.
>
> ---
>
> >*Q2: What does “low-level neural responses” in introduction section refer to? Does it mean the low-level components within the neural signals themselves, or the neural responses elicited by low-level visual stimuli?*
>
> *Response to Q2*: Thanks for your question.
>
> "Low-level neural responses" refer to the neural responses elicited by low-level visual stimuli.
>
> ---
>
> >*Q3: What’s the meaning of E_d in Equation 1. Besides, the parentheses in Equation 1 are mismatched.*
>
> *Response to Q3*: Thanks for your question. We revise "the contributions of $E_d$" on line 217 to "the contributions of edge feature $E_d$". Additionally, we correct the mismatched parentheses in Equation (1).
>
> ---
>
> >*Q4: Equation (5) seems to contain a possible typo: the image representation should be denoted as $v$ rather than $t$.*
>
> *Response to Q4*: Thanks for your careful review. We **correct the spelling error in Equation (5)**, the symbol should be '$v$' instead of '$t$'. We also check the relevant descriptions that reference this equation to ensure the consistency of all symbols.
>
> We hope these responses adequately address your concerns, and we remain open to any further discussion.
>
> ---
>
> [1] Marr D, Hildreth E. Theory of edge detection. Proceedings of the Royal Society of London. Series B. Biological Sciences 1980.
>
> [2] Choi M, et al. A dual-stream neural network explains the functional segregation of dorsal and ventral visual pathways in human brains. NeurIPS 2023.

---

### Official Review · Reviewer_pyrJ · 2025-10-31

**Soundness:** 2
**Presentation:** 2
**Contribution:** 1
**Rating:** 2
**Confidence:** 4

**Summary:**

This paper focuses on the cross-modal retrieval task between EEG signals and images. The authors propose their method based on several, somewhat scattered motivations, which they summarize as follows:

1. Existing deep models do not align well with human visual mechanisms;

2. The modality alignment strategy in CLIP overlooks the correlations among unpaired data;

3. The human visual mechanism is dynamic and bidirectional, while existing methods are static and unidirectional (which is actually a subset of motivation 1).

At the methods level, the authors introduce three main designs:

1. Based on the low-level visual features of images, they compute a pixel-wise mask or attention matrix;

2. They design a semantic-guided alignment loss (this approach has already been mentioned in other related works—see the Limitation section);

3. They propose a dynamic loss weighting strategy.

The authors evaluate their method on two brain–image datasets for the cross-modal retrieval task.

In my opinion, the contribution of this paper to the field is limited and does not introduce much that is genuinely novel.

**Strengths:**

1. The manuscript is well-structured with clear formatting.

**Weaknesses:**

1. The motivation of this paper is not sufficiently in-depth. In my view, the authors merely list some common issues—for example, that deep networks do not align with human visual mechanisms, and that CLIP’s cross-modal alignment ignores the correlations among unpaired data. These motivations are applicable to many tasks and are not specific to the EEG–image retrieval task.

2. The semantic-guided alignment strategy proposed by the authors in Section 3.3 has already been introduced and applied in previous brain–image retrieval studies [1]. The authors should properly cite these previous works and discuss the similarities and differences.

3. The experimental evaluation is not sufficient. The authors only conducted evaluations on the retrieval task, whereas many recent studies based on the Things-EEG and Things-MEG datasets have also performed evaluations on image reconstruction tasks.




[1] Qiongyi Zhou et al. CLIP-MUSED: CLIP-guided multi-subject visual neural information semantic decoding. ICLR 2024.

**Questions:**

1. In line 82, the authors mention that deep learning–based visual encoders extract high-level visual features, which leads to misalignment. However, I notice that even though the authors compute a pixel-wise attention map based on low-level visual features to obtain 𝐼_𝑎, they still feed 𝐼_𝑎 into such a visual encoder. This approach does not seem to address the misalignment problem, as it still relies on extracting high-level visual features?

2. I am curious about what the computed pixel-wise attention 𝑅_𝑎 looks like. Could the authors provide a visualization of it? Additionally, the subsequently computed 𝐼_𝑎 might also benefit from being visualized.

3. I don’t quite understand how the proposed dynamic loss weighting strategy can address the bottleneck 3 mentioned by the authors.

4. I couldn’t seem to find the definition of 𝐸_𝑑 in line 217.

5. Some citation formats need to be corrected, for example, in line 262.

---

> ### Author Response · Authors · 2025-11-27
> **Response to Reviewer pyrJ Part (1/2)**
>
> Thanks for the reviewer's insightful feedback and helpful suggestions. We highlight the corresponding revisions in **blue** for clarity in the revised paper. We will answer your questions point by point below.
>
> ---
>
> >*W1: The motivation of this paper is not sufficiently in-depth...These motivations are applicable to many tasks and are not specific to the EEG–image retrieval task.*
>
> *Response to Summary and W1*:
>
> Thank you for your questions, and we apologize for any confusion caused by our original description.
>
> Our motivation stems from a core challenge in EEG-image retrieval: the **feature misalignment** between low-SNR EEG and visual data. Our technical innovations directly address this by simulating the brain's dual-stream processing across three key stages:
>
> (1) **Resolving Input Misalignment**: our bottom-up pathway first establishes a stable **perceptual grounding** by aligning EEG with salient visual features, directly mitigating the low-level feature mismatch between the two modalities.
>
> (2) **Leveraging Semantic Priors**: simply aligning EEG with CLIP's image embeddings is insufficient for noisy neural signals. So we employ **KL divergence** to align the (Image-EEG) distribution with the more structured (Image-Text) distribution, enabling the model to learn fine-grained semantic representations.
>
> (3) **Neuroscience-Inspired Dynamic Regulation**: DLA module **coordinates these two pathways** by prioritizing perceptual grounding before engaging in semantic alignment. This approach enhances biological plausibility by simulating how priors regulate perception, and ensures a stable and effective learning process.
>
> ---
>
> > *W2: The semantic-guided alignment strategy proposed by the authors in Section 3.3 has already been introduced and applied in previous brain–image retrieval studies...*
>
> *Response to W2*: Thank you for this question.
>
> We thank the reviewer for highlighting CLIP-MUSED [1]. We have now discussed **in the revised Section 2.2 (line 160)** .
>
> **The content cited in our paper** is : "Similar applications also exist for high-spatial-resolution fMRI, which relies on Representational Similarity Matrices (RSM) to align its global topology (Zhou et al., 2024)."
>
> ---
>
> > *W3: The experimental evaluation is not sufficient...*
>
> *Response to W3*: Thanks for your suggestion.
>
> We agree that image reconstruction is an important task in neural decoding. But this work's core task is semantic alignment, for which **retrieval serves as a direct evaluation metric**. To ensure a consistent benchmark, our comparison includes six leading methods specifically proposed for retrieval.
>
> Inspired by this suggestion, we **add a discussion in our conclusion (line 538)** on extending NeuroAlign to generation tasks as future work.

---

> ### Author Response · Authors · 2025-11-27
> **Response to Reviewer pyrJ Part (2/2)**
>
> > *Q1: …I notice that even though the authors compute a pixel-wise attention map based on low-level visual features to obtain 𝐼_𝑎, they still feed 𝐼_𝑎 into such a visual encoder. This approach does not seem to address the misalignment problem…*
>
> *Response to Q1*: Thank you for this question.
>
> (a) We use low-level features as an **attentional guide** for the subsequent extraction of high-level features, **rather than replacing** them entirely. Neuroscience indicates that robust recognition requires integrating bottom-up inputs with top-down priors, as low-level features alone are insufficient [2-4].
>
> (b) We provides detailed ablation experiments in Section 4.2.4 Ablation Study (line 486) and Appendix A.3.5 Ablation Experiments (line 1026), quantifying the performance improvement of using $I_a$ compared to directly using $I$.
>
> (c) We show the **results in the table below**, which demonstrate that VSE effectively mitigates feature misalignment and improves the accuracy of brain signal image retrieval.
>
> | Dataset       | Intra-subject (Top-1)      | Intra-subject (Top-5)      | Inter-subject (Top-1)     | Inter-subject (Top-5)     |
> | :------------ | :------------------------: | :------------------------: | :-----------------------: | :-----------------------: |
> | **THINGS-EEG2** | **48.1** / 40.1 (↓8.0) | **78.1** / 73.3 (↓4.8) | **14.5** / 12.3 (↓2.2) | **36.4** / 33.5 (↓2.9) |
> | **THINGS-MEG**  | **26.5** / 21.1 (↓5.4) | **53.3** / 47.6 (↓5.7) | **3.9** / 3.1 (↓0.8)   | **13.1** / 11.0 (↓2.1) |
>
> ---
>
> > *Q2: I am curious about what the computed pixel-wise attention 𝑅_𝑎 looks like. Could the authors provide a visualization of it? Additionally, the subsequently computed 𝐼_𝑎 might also benefit from being visualized.*
>
> *Response to Q2*: Thank you for this question.
>
> The pixel-wise attention map R_a and the attention-weighted image I_a are **visualized in Appendix A.2.1 (Fig.7, line 830)**, corresponding to the Fig.7(b) and Fig.7(e), respectively.
>
> To better understand, we have revised the description and caption about Figure 7, clarifying the specific function of each module.
>
> ---
>
> > *Q3: I don’t quite understand how the proposed dynamic loss weighting strategy can address the bottleneck 3 mentioned by the authors.*
>
> *Response to Q3*: Thank you for this question, and we apologize for any confusion arising from our original description. We revise the paper (line 299) to better explain how the DLA mechanism directly addresses Bottleneck 3.
>
> Our DLA mechanism resolves Bottleneck 3 by making the loss weights adaptive, modeling the weights $\mathcal{w}^i_t(g^i_t, r^i_t), i \in [1,2,3]$ as a function of time $t$, gradient magnitude $g_t^k$, and rate of change $r_t^k$. This simulates the dynamic learning process of semantic information over time, trends, and instantaneous changes in the brain's 'bottom-up' and 'top-down' dynamic learning processes.
>
> **Initially**, DLA assigns a high weight to the direct EEG-Image alignment loss ($L^1$), forcing the model to first focus on aligning low-level, stimulus-driven features. **As this initial alignment stabilizes**, DLA responsively increases the weight of the semantic guidance loss ($L^3$), shifting the model's focus toward aligning high-level semantic concepts. This dynamic process effectively mimics the brain's cognitive shift from a stimulus-driven response to semantically-guided processing, resulting in a more efficient alignment.
>
> ---
>
> > *Q4: I couldn’t seem to find the definition of $𝐸_𝑑$ in line 217.*
>
> *Response to Q4*: Thanks for your question.
>
> We revise "the contributions of E_d" on line 217 to "the contributions of edge feature E_d", where **E_d represents the edge feature map** generated by the Rapid Saliency Detection module.
>
> ---
>
> > *Q5: Some citation formats need to be corrected, for example, in line 262.*
>
> *Response to Q5*: Thank you for pointing out these details.
>
> We correct the citation format issue on line 262 and carefully reviewed all the citation formats throughout the entire paper. All relevant revisions have been completed in the revised version.
>
> We hope our responses clarify the reviewer’s concerns, and we welcome any further discussion or questions.
>
> ---
>
> [1] Qiongyi Zhou, et al. CLIP-MUSED: CLIP-guided multi-subject visual neural information semantic decoding. ICLR 2024.
>
> [2] Kar K, et al. Evidence that recurrent circuits are critical to the ventral stream’s execution of core object recognition behavior. Nature neuroscience 2019.
>
> [3] Bar M, et al. Top-down facilitation of visual recognition. Proceedings of the national academy of sciences 2006.
>
> [4] Chiou R, et al. The anterior temporal cortex is a primary semantic source of top-down influences on object recognition. Cortex 2016.

---

### Author Response · Authors · 2025-12-03
**Summary of Contribution and Rebuttal**

Dear ACs,

We sincerely thank you for taking the time and effort to handle our submission. We are deeply grateful to all reviewers for their thoughtful and constructive comments, which are invaluable in improving our paper.

We are encouraged that the reviewers recognized the strengths of our work.

- Reviewer pyrJ notes that “the manuscript is well-structured with clear formatting”.

- Reviewer A8NT considers that “the paper is well-written, with three motivations clearly illustrated”, and “the experiments demonstrate the effectiveness over the baseline model”.

- Reviewer LVmN emphasizes that our method is “clear and addresses specific bottlenecks”, “the inter-subject improvement is statistically meaningful given the difficulty of EEG generalization” and “the analysis convincingly demonstrates alignment”.

At the same time, we take the reviewers' suggestions seriously and further refine our work.

Below, we briefly summarize our main contributions and the improvements made in our rebuttal:

- We mitigate the feature mismatch by grounding noisy neural signals in stable visual saliency features via a saliency extraction module.

- We address the semantic inconsistency by employing KL divergence to align neural signals with structured Image-Text distributions.

- We enhance stability and biological plausibility via a dynamic mechanism that mimics the brain’s perceptual-to-semantic progression.

We comprehensively address all reviewers' concerns, including biological plausibility, mechanism interpretability, and experimental validation. Following these revisions, **Reviewer LVmN raised the score to 8 strictly under double-blind conditions.** Regarding motivation, while Reviewers A8NT and LVmN explicitly validated its clarity, we further sharpen our arguments on EEG-specific constraints in response to Reviewer pyrJ’s concern. The remaining comments requested clarifications or additional analyses, which are addressed point by point in the rebuttal.

We believe this new perspective opens a promising direction for bridging the gap between noisy neural signals and visual semantics, while simultaneously improving performance and adhering to biological plausibility.

In light of these contributions, we sincerely hope you will consider supporting our submission.

Best regards,

Authors

---

### Meta-Review · Area_Chair_eY9s · 2026-01-12

**Summary:**

The paper presents NeuroAlign, a framework for EEG-image retrieval that mimics the brain’s dual-stream visual processing through saliency extraction, semantic guidance via LLMs, and dynamic loss weighting. While the reviewers found the manuscript well-structured and acknowledged the performance gains on the THINGS datasets, the overall consensus leans toward rejection due to concerns regarding novelty and methodological depth.

Specifically, Reviewer pyrJ noted that the motivations are somewhat generic and that the semantic-guided alignment strategy closely resembles prior work such as CLIP-MUSED, limiting the paper's original contribution. Furthermore, Reviewer A8NT highlighted a conceptual contradiction: the authors aim to ground the model in low-level features to avoid overfitting, yet continue to rely on high-level visual encoders, potentially undermining the primary goal of resolving feature misalignment. Additionally, the experimental evaluation was criticized for being limited to retrieval, omitting the image reconstruction benchmarks common in recent neural decoding literature.

**Reviewer Concerns:**

See summary above.

**Reviewer Scores:**

It is hard to predict. The authors did a good job in addressing the reviewers' questions.

---

### Decision · Program_Chairs · 2026-01-26

Reject